# Individual traits and experiences predict the content of dreams
Valentina Elce [1] ✉, Giorgia Bontempi [1], Serena Scarpelli[2], Bianca Pedreschi[1], Pietro Pietrini[1], Luigi De Gennaro[2], Michele Bellesi [3], Giulio Bernardi [1,4] ✉ & Giacomo Handjaras [1,4] ✉

Dreams are universal yet highly idiosyncratic experiences. While memories and personal concerns are known to influence dream content, how such influences evolve over time and how stable individual traits shape dreaming remain unclear. Here, we systematically quantified the semantic structure of dreams in a large, multimodal dataset comprising 3366 reports of dreams and waking experiences collected from 207 adults between 2020 and 2024, alongside demographic, cognitive, psychometric, and sleep measures. To this end, we combined large language model-assisted evaluation of hypothesis-driven semantic dimensions and a data-driven lexical domain approach. Relative to waking reports, dreams shifted from self-referential, thought-centered narratives to perceptual experiences dominated by visuo-spatial details, multiple characters, and bizarre events. Stable traits, including attitude toward dreaming, mind-wandering propensity, and subjective sleep quality, selectively influenced dream content. A second, independent dataset collected during the first 2020 COVID-19 lockdown (80 participants) allowed us to examine the impact of a major external stressor on dream semantics. During lockdown, dreams showed increased references to limitations and heightened emotional intensity, effects that gradually normalized over the following years. These findings demonstrate that stable individual traits and incidental experiences jointly shape dream semantics.

Dreams are both universal and deeply personal experiences, intimately connected to waking life, yet often strikingly distinct from it. When we sleep, our brain generates a diverse array of immersive experiences—sometimes mundane, sometimes surprising, and occasionally delightful or terrifying. While the artistic, cultural, and psychological significance of dreaming is undeniable, dreams have also gained increasing attention as a subject of rigorous neuroscientific study in recent decades[1–3]. Indeed, dreams have been proposed not only as a unique model for studying the emergence of consciousness in the human brain but also as a potential window into the functions of sleep itself[1,4]. As subjective conscious experiences arising during sleep, dreams emerge when the brain is at least partially disengaged from external reality, unfolding as dynamic narratives shaped by prior waking experiences, personal beliefs and expectations, and neural mechanisms that are only beginning to be fully understood[5,6].

The analysis of dream content following experimentally controlled manipulations of individuals' experiences has led to the hypothesis that dreams, rather than being a mere byproduct of neural activity during sleep,

may serve an evolutionary function, related to the mechanisms of learning[7], memory consolidation[8], or emotion regulation[9]. Consistent with this view, numerous studies report continuity between waking concerns and dream experiences: characters, places, and situations encountered while awake often reappear in transformed form during dreams[10,11], and waking emotions, particularly negative affect and anxiety-related traits, are associated with more negative dream content and, in some cases, nightmares[12,13]. Collectively, these observations support the idea that dreaming may aid the individual to cope with waking life events, understand them, and learn from them[14,15].

At the same time, dreams do not simply reproduce waking life: they tend to integrate waking elements in metaphorical or associative ways that vary with each dreamer's history, beliefs, and concerns. Although common themes can emerge across individuals[16], dreams are typically regarded as highly variable and idiosyncratic, and it remains unclear to what extent stable interindividual differences shape what people dream about. While a large literature has examined how individual factors relate to dream recall, far fewer studies have linked stable traits (e.g., personality, cognitive style) to

[1]MoMiLab Research Unit, IMT School for Advanced Studies Lucca, Lucca, Italy. [2]Department of Psychology, Sapienza University of Rome, Rome, Italy. [3]School of Biosciences and Veterinary Medicine, University of Camerino, Camerino, Italy. [4]These authors contributed equally: Giulio Bernardi, Giacomo Handjaras. ✉e-mail: valentina.elce@imtlucca.it; giulio.bernardi@imtlucca.it; giacomo.handjaras@imtlucca.it

dream content—its perceptual richness, themes, emotional tone, or bizarreness—and those that did often relied on small samples, narrow predictor sets (commonly age[17], sex[18], or affective traits[19]), and retrospective questionnaires. Moreover, while laboratory paradigms demonstrated that pre-sleep experiences can modulate dreams[20], these manipulations are typically brief, artificial, and of variable personal relevance, limiting inferences about long-term, real-world dynamics. Therefore, it remains uncertain whether shared external events may influence dream content at the population level, and how such influences may evolve across long time periods such as months or years.

Taken together, the available evidence depicts dreaming as characterized by both continuity and discontinuity with waking life. As such, it remains unclear to what extent stable individual traits and intervening experiences independently and interactively contribute to shaping dream content and its subjective qualities. Several theoretical accounts of dream generation assign a privileged role to specific cognitive traits, such as the propensity for mind wandering[21], often conceptualized as a waking analogue of dreaming, or the vividness of visual imagery during wakefulness[22]. However, direct empirical tests of these hypotheses within comprehensive models that simultaneously account for multiple interacting predictors and potential confounds remain scarce. Clarifying how enduring individual characteristics and shared environmental influences jointly contribute to dream construction is therefore essential for advancing mechanistic models of dreaming.

To address these gaps, we collected a large, prospective multimodal dataset including first-person reports of both dream and wakefulness experiences, and applied validated natural language processing (NLP) tools to quantify their semantic features. We combined large language model (LLM)-assisted ratings of hypothesis-driven semantic dimensions with a data-driven, lexical-domain approach capturing fine-grained content categories. While classical manual coding systems typically assess dream content in a categorical or presence/absence format, our NLP-based approach enabled a finer-grained, quantitative assessment of the same constructs on continuous scales, capturing the relative degree to which each semantic feature was expressed in the reports.

We pursued three aims. First, we quantified similarities and differences between dream and waking reports, anticipating both broad continuities and principled discontinuities in semantic features. Second, we tested whether stable individual traits—including cognitive abilities, personality, imagery, and sleep-related characteristics—systematically relate to dream content and structure beyond recall frequency, with particular attention to factors that differentially influence dreaming versus wakefulness. Third, leveraging multi-year data spanning the COVID-19 period and the subsequent four years, we examined whether the pandemic period, representing a shared stress event, has been associated with population-level changes in dream semantics and whether potential alterations subsequently varied over time.

Our results provide a reproducible, scalable framework that bridges individual idiosyncrasy with generalizable patterns of dream formation, moving beyond small, retrospective, theme-limited studies toward a quantitative map of dream content comparable across individuals, datasets, and time.

## Methods
### Datasets and overall design
We employed NLP techniques to quantify the semantic features of verbal reports of wakefulness and dream experiences from two distinct prospective datasets. The *main* dataset, collected between March 2020 and March 2024, included reports from the general Italian population and was used to examine dream content under physiological conditions while identifying the influence of trait and state variables. The *lockdown* dataset, collected during the COVID-19 lockdown (April–May 2020) was analyzed to explore the potential impact of external large-scale stressors on dream content. In both studies, participants completed a two-week prospective dream-diary protocol providing daily reports immediately after awakening. Participants

in the *main* dataset additionally underwent a comprehensive assessment of waking experiences, psychological and cognitive traits, and sleep patterns to enable multimodal characterization of individual differences. In particular, they completed three distinct phases: (i) an initial screening interview followed by the completion of a series of questionnaires, (ii) an experimental period lasting 15 days, during which sleep patterns and verbal reports of subjective conscious experiences during sleep and wakefulness were recorded, and (iii) a concluding session involving an exhaustive cognitive assessment.

Furthermore, two control experiments were performed to evaluate the reliability of the automated methods used to quantify the semantic content of subjective experiences. In the first study, four trained external human raters manually scored a subset of dreams from the *main* dataset. In the second study—an independent dream-diary protocol—participants recorded and subsequently evaluated the content of their own dreams. Together, these experiments assessed how accurately the automated ratings captured both external evaluations and participants' subjective experiences of their dreams. These experiments and their results are detailed in Supplementary Text and Supplementary Figs. S1 and S2.

### Participants
The *main* dataset was collected from an initial sample of 217 Italian native language speakers. Participants were drawn from the central and northern regions of Italy and their recruitment was conducted through word-of-mouth and flyer distribution. Inclusion criteria were: age from 18 to 70 years old; having regular sleep/wake patterns with six to eight hours of sleep per night, and no diagnosis of sleep-related, neurological, or psychiatric disorders. Exclusion criteria included taking medications that could have affected sleep patterns and having a recent (last 6 months) history of alcohol and drug abuse. Moreover, women who were pregnant, were planning a pregnancy, or were breastfeeding at the time of the study were excluded. Recruitment achieved a balanced distribution of biological sex and encompassed a wide range of ages and educational backgrounds. Ten participants failed to comply with the experimental protocol and dropped out from the study, leading to a final sample of 207 participants (117 female participants, 90 male participants; mean ± std, age 34.9 ± 12.4 yrs, range 18–69 yrs; years of education 16.8 ± 2.8 yrs, range 8–23 yrs). As detailed below, participants collectively provided 1687 dream reports and 2843 waking reports. The study was conducted under a protocol approved by the Local Joint Ethical Committee for Research (#11/2020). All volunteers signed a written informed consent form before taking part in the study and retained the faculty to drop from the study at any time. No financial compensation was offered for participation.

The *lockdown* dataset was collected from an initial sample of 100 volunteers recruited via social media. Only adults residing in Italy with regular sleep/wake patterns and no history of sleep-related, neurological, or psychiatric disorders were included. Those on medications affecting sleep were excluded. Volunteers first completed an online survey collecting sociodemographic data and assessing eligibility. Sex, age, and education were self-reported. Participants were instructed to keep a dream diary from April 28 to May 11, 2020. The first week (April 28–May 4) corresponded to strict lockdown, while the second (May 5–May 11) marked the easing of restrictions. Of the initial 100 participants, 80 (60 female participants, 20 male participants, age 25.6 ± 4.1 yrs, age range: 19–41 yrs, years of education 14.8 ± 2.1 yrs, range 13–18 yrs) completed the study[23]. The dream-diary protocol yielded a total of 351 dream reports. Informed consent was obtained, and participants could withdraw at any time. No financial compensation was provided. The study was approved by the Institutional Review Board of the Department of Psychology at Sapienza University of Rome (#0000646/2020).

Information on participants' socioeconomic status, race, ethnicity, or communities of descent was not collected. The analysis plan for this study was not preregistered. Unless otherwise specified, the following sections refer to the experimental protocol, data collection, and data-processing procedures applied to the *main* dataset.

## Sample characterization

Participants underwent an anamnestic interview aimed at assessing their general health and adherence to inclusion/exclusion criteria. Biological sex, age, and education were determined by self-report. Recruited participants were then asked to fill out several questionnaires aimed at investigating their attitude towards dreaming[24], trait anxiety levels (*State-Trait Anxiety Inventory, STAI*; ref. [25]), vividness of visual imagery (*Vividness of Visual Imagery Questionnaire, VVIQ*; ref. [26]), proneness to mind-wandering (*Mind Wandering - Spontaneous and Deliberate Scale; MW*; ref. [27]), sleep quality (*Pittsburgh Sleep Quality Index, PSQI*; ref. [28]), chronotype (*Morningness-Eveningness Questionnaire; MEQ*; ref. [29]). Attitude towards dreaming was assessed using a six-item questionnaire where participants were asked to provide their degree of agreement with six statements regarding the general meaning and significance of dreams on a Likert Scale from zero ('completely disagree') to four ('completely agree'). Three items were positive statements about dreams (e.g., 'dreams are a good way of learning about my true feelings') and three were negative (e.g., 'dreams are random nonsense from the brain'). A global score was computed by subtracting the sum of scores provided to the negative statements from the sum of scores associated with the positive statements. Finally, participants evaluated their daytime sleepiness (*Epworth Sleepiness Scale, ESS*[30]) and completed a dream questionnaire extended and adapted from Schredl and colleagues[31] concerning their perceived dream recall frequency and several aspects of the dream experiences of the previous three months, including, among others, vividness, bizarreness, and valence. These last two questionnaires were not included in the analyses described in this study.

At the end of the 15-day period, during the final visit, all participants underwent a neuropsychological assessment lasting one hour and aimed to evaluate their cognitive status and to account for potential age-related differences in cognitive functioning (age range: 18–70 yrs). The battery included: *Mini-Mental State Examination*, for a general screening of participants' cognitive abilities[32]; *Stroop Task*, for assessing participants' vulnerability to cognitive interference[33]; *Babcock Story Recall Test*, for evaluating participants' verbal memory[34]; *Rey–Osterrieth complex figure*, for visuospatial memory[35]; *Phonemic and Semantic Fluency* for evaluating participants' semantic memory[36]; *Token Test* focusing on syntactic abilities in language[37]; *Wechsler Adult Intelligence Scale* (*WAIS*) - *vocabulary subtest*[38], yielding information regarding individuals' semantic knowledge and verbal production and comprehension abilities. Finally, participants were required to complete a free picture description task, extracted from the *Battery for the analysis of aphasic deficits (Batteria per l'analisi dei deficit afasici, BADA*[39]), aimed at quantitatively evaluating participants' connected speech abilities and verbosity by providing them with two pictures representing complex scenes that they had to describe, giving as many details as possible, without any time limit. To obtain a quantitative measure of participants' verbosity, we computed the word count for the descriptions of both images, averaged the results, and then computed the logarithm of this mean. The complete administration of all neuropsychological tests required about one hour. Among these, the following tests were included in the analyses: *Stroop Task (cognitive interference), Babcock Story Recall Test (verbal memory), Rey–Osterrieth complex figure (visuospatial memory)*, and the verbosity measure extracted from the *BADA* free description task.

Importantly, the questionnaires and cognitive tests were selected to provide a comprehensive profile of individual characteristics known or hypothesized to influence dreaming. Previous research on dream recall frequency, a related but more extensively studied phenomenon, has shown that several stable traits and cognitive abilities modulate the frequency and richness of dream reports[40–42]. Building on this evidence, we sought to extend these insights to dream content, which remains comparatively underexplored. Specifically, we included measures capturing imagery abilities, as individual differences in visual imagery have been associated with the vividness and perceptual detail of dreams[43–45]; attitude toward dreaming, reflecting the motivational and metacognitive factors known to enhance dream recall[24,41]; and mind-wandering proneness, given the proposed continuity between waking spontaneous thought and dreaming[10,46]. We also

included memory and executive functions, which may influence how experiences are encoded and reconfigured during sleep. Sleep quality and chronotype were evaluated to account for sleep-related influences on dream phenomenology[47,48]. Finally, trait anxiety was included based on evidence that anxious and negative affective states in wakefulness are associated with more negative or emotionally intense dream content[2,49,50]. Together, these instruments were intended to capture a multidimensional profile of cognitive, emotional, and sleep-related traits that could systematically shape the structure and semantics of dream content.

## Recording of verbal dream reports and sleep patterns

Participants were provided with an actigraph and a voice-recorder and were asked to record each morning, upon awakening from sleep, everything that was going through their mind just before they woke up, everything they remembered, every experience or thought they had before awakening. Participants were asked to specifically focus on the very last experience they had before the morning awakening in order to minimize the effect of confounding factors that may interfere between the dream experience and its retrieval. Regardless of whether or not they might remember the content of their sleep experiences, participants were required to provide a report. In the event that they woke up with the perception of having been dreaming but could not recall any feature of the experience (i.e., "*white dream*"[51]), they were asked to describe this feeling. Similarly, if they woke up with the perception of not having been dreaming at all before waking up, they were asked to report this. Although participants were explicitly instructed to report only the experiences remembered immediately upon awakening, some occasionally recalled and recorded their dreams later in the day. Since these memories could be influenced and altered by external stimuli and waking experiences, we opted to exclude those data from the current analysis (less than 0.1%). After excluding morning reports without descriptions of contentful experiences, the final sample consisted of 1687 dream reports (reports per participant: 8.1 ± 3.6, min:1, max:16).

Moreover, during the 15 days of the study, participants wore an actigraph to track sleep-wake patterns (MotionWatch-8, Camtech). A subgroup of 50 volunteers (27 female participants, 23 male participants; age 29.7 ± 5.2 yrs, range 22–44 yrs) also had their sleep-related brain activity recorded through a portable Electroencephalographic (EEG) system (*DREEM*) equipped with five EEG dry electrodes (seven derivations: Fp1-O1, Fp1-O2, Fp1-F7, F8-F7, F7-O1, F8-O2, Fp1-F8), a pulse sensor, and a 3D accelerometer. Eight participants interrupted EEG data collection due to discomfort while sleeping. Therefore, we were able to collect neural activity from 42 participants. EEG data were not included in the analyses described in this study.

## Recording of verbal wakefulness reports

With the aim of measuring semantic content differences between wakefulness and dreams, participants were prompted at pseudo-random times throughout the day, once per day, to record the very last experience that was going through their minds up to 15 min before. They received a simple text message containing the word "record" ("*registra*") as a prompt. This approach minimized reliance on long-term memory by capturing thoughts close to their occurrence[52]. Overall, after excluding promptings which were not followed by a recording (*n* = 238), we collected 2843 wakefulness reports. In this analysis, to account for the potential impact of recent waking experiences in dreams, we downsampled each participant's wakefulness reports to match the dream reports. Specifically, we selected wakefulness reports from the experimental days preceding recorded dreams; if unavailable, we randomly selected unassigned reports (if available) as substitutes. The random sampling accounted for about 13% of the selected wakefulness reports. The final sample consisted of 1679 wakefulness reports (reports per participant: 8.1 ± 3.5, min:1, max:16).

## Verbal report management and preprocessing

The recordings of participants' verbal reports were automatically transcribed by means of the Microsoft 365 Word transcriber (Microsoft

Corporation). Afterwards, textual data underwent a three-level preprocessing procedure.

At the first level, experimenters manually verified the correspondence between speech and the transcriptions. When needed, they corrected the recordings and added punctuation marks according to the rules of Italian grammar. Verbal reports were also anonymized in order to protect any sensitive data provided by the participants while describing their experiences: any information that would have allowed to identify the volunteers (i.e., proper nouns of cities, places, people) was replaced by a code (e.g., proper nouns of people were replaced by the code [NAME1], [NAME2], etc.).

Afterwards, a second level preprocessing was performed in order to remove any lexical item that did not directly refer to the experiences. In line with previous studies[42,53], textual data were manually pruned of volunteers' commentary about the night (e.g., «Tonight I slept badly»), about the task upon awakening and during the day (e.g., «I remember very few things», «What was I thinking before you sent me the message?»), about the experience itself (e.g., «That doesn't make any sense», «I don't know if this is interesting for you»); linguistic expressions used to introduce the experience (e.g., «Goodmorning, tonight I dreamt of…» or, in the case of wakefulness reports, «Before receiving the message…»); and false starts, repetitions and self corrections (e.g., «Kin… kind of of like a boat»). We also converted arabic numbers into numerals (e.g., the number "3" was converted into the word «three»), corrected misspelled words, and regionalisms and dialectal words (e.g., «and I asked him 'how ya doin'?'») were replaced with the Standard Italian equivalent. Afterwards, UTF-8 encoded textual data were fed into three AI models for semantic feature evaluation, as described in the section below.

Moreover, at third-level preprocessing, grammatical categories or parts-of-speech (POS; i.e., categories including, among others, nouns, verbs, adjectives, adverbs, interjections, and conjunctions[54]) were assigned to each word occurring within the verbal reports by means of the *Stanza* toolkit[54]. *Stanza* provides state-of-the-art accuracy for Italian POS tagging, with reported error rates below 1%, which were confirmed through manual verification on a random subsample of our data. Reports and POS tagged texts were then processed, and statistical analyses were performed using MATLAB (The Mathworks Inc., 2024b) as described below.

## AI scoring of semantic dimensions

Second-level preprocessed textual data were fed into three LLMs: LLaMA 3[55], ChatGPT-4, and ChatGPT-4 Turbo[56], by using a chat prompting pipeline[57,58]. The models evaluated each verbal report across 16 hypothesis-driven semantic dimensions, designed to capture key experiential and structural aspects of dream and waking experiences, and rated each dimension on a 9-point Likert scale[59]. The selected dimensions were informed by both classical approaches to dream content analysis (particularly manual coding systems such as the Hall and Van de Castle scale) and by the specific aims of the present study[60,61]. We sought to capture a broad range of experiential components encompassing perceptual and cognitive features, affective dimensions, character-related interactions, and contextual aspects of the experience. In addition, one dimension specifically assessed the incorporation of experimental procedures, a phenomenon frequently observed in laboratory-based dream studies[62] but less characterized in home settings. Including this dimension allowed us to evaluate the extent to which participation in the study itself may have influenced dream content. Finally, the selected dimensions were chosen to ensure reliable identification through LLM-based semantic evaluation, maximizing interpretability and consistency across models. The models were prompted based on the following definitions (for the specific instructions provided, see Supplementary Table S1):

– *Incorporation of experimental procedures*[62], that is reporting—either in wakefulness or dream reports—any reference to the experimental protocol the volunteers were asked to perform when participating in the current study (e.g., dreaming of waking up and recording a dream, pressing the event marker of the actigraph, wearing the portable EEG, talking with one of the experimenters or any other character about the study);

– *Thought*, referring to the presence of abstracts thoughts or reasoning of the narrator;

– *Visual experiences*, that is any explicit reference to the sight, but also to details that can be only perceived through it and actions requiring vision, such as looking for somebody in a crowd or aiming at a target;

– *Auditory experiences*, including any reference to the hearing, to details that could be only perceived with it, such as music, noise or voices, and actions that would imply the hearing;

– *Tactile experiences*, meaning not only explicit references to the touch, but also to actions implying the touch, such as brushing, caressing or grasping something with any part of the body, and references to details which can be perceived only with the touch;

– *Valence*, meaning the degree of pleasantness or unpleasantness of the emotional tone of dream and wakefulness reports, irrespective of the emotional intensity, and ranging from extremely negative (e.g., in the case of reports holding feelings of sadness or anger or despair) to extremely positive (e.g., reports carrying feelings instead of happiness, joy or excitement);

– *Arousal*, that is the emotional intensity or strength of reports, regardless of their valence;

– *Bizarreness*, meaning the degree of illogicality, strangeness or discrepancy[63] of the events described within the reports;

– *Social processes*, that is any explicit or implicit reference to social interactions, such as talking or arguing with somebody or participating in an activity with other people;

– *Movement*, meaning actions performed by the narrator or any other character within the report that implied movements of the body (e.g., running, jumping, swimming);

– *Space*, that is any reference to details regarding the environment where the events took place (e.g., descriptions of buildings, rooms or landscapes);

– *Change of settings*, including not only explicit references to the displacement of objects or characters from one setting or location to another but also to sudden changes of setting without the explicit reference to an actual movement;

– *Time*, that is references to the chronological aspects of the events, as well as to details regarding their duration;

– *Body*, meaning references to any body part and bodily function or instinct, such as eating, drinking, sleeping or having sex;

– *Limitations of freedom*, meaning elements that restricted or could restrict characters' freedom of action within the narratives, both physically (e.g., a blocked road or the feeling of wanting to move but being blocked by an invisible force) and in terms of social or moral norms (e.g., refraining from doing something because deemed inappropriate in that particular situation);

– *Agentivity*, assessing how much the narrator actively performed the actions described within the events rather than passively undergoing them.

We computed the median across the scores of the three AIs, obtaining one value for each report and dimension. We also measured the reliability of AI scores in two control experiments (see Supplementary Figs. S1 and S2).

## Lexical domain analysis

To complement the hypothesis-driven analysis of semantic dimensions, we implemented a data-driven lexical domain approach aimed at identifying references to distinct semantic elements emerging from the reports themselves. This analysis was designed to quantify the semantic content at the lexical level without imposing pre-defined topical categories. We first constructed a bag-of-words representation and generated word embeddings to model the semantic space of all lexical items appearing across dream and wakefulness reports. This high-dimensional semantic space was then

**Article**

decomposed into coherent lexical domains based on the clustering of semantically related words, yielding domain embeddings that captured recurring conceptual groupings within the corpus. For each dataset, lemmas were extracted and their cosine similarity to each domain embedding was computed, resulting in a normalized lemma-by-domain matrix. A weighted sum of all lemma scores per report was then calculated to produce a continuous report-by-domain matrix. Finally, this matrix was binarized using a set of null reports to establish sensitivity- and specificity-based thresholds, ensuring that only domains showing meaningful semantic representation were retained. A detailed, step-by-step description of the procedures applied for lexical domain generation and analysis is provided in the following sections.

### Identification of the domains

Given the high variability in semantic content, length, and the lack of clear polarization toward specific topics in verbal reports, rather than extracting topics directly from the documents (as done in region models[64]), we decomposed the semantic space at lemma-level into a set of domains, each centered around a specific topic.

We first identified all lexical items across verbal reports from the *main* dataset using POS tagging implemented in the *Stanza* toolkit. We restricted the selection to pronouns and semantically meaningful words: nouns, verbs, adjectives, and adverbs, hence excluding words tagged as conjunctions, determiners, and interjections, that is not entailing a lexical meaning but rather a grammatical function[54]. To obtain numerical vectors for each lexical item, we relied on pre-trained word-embeddings obtained from Wikipedia (snapshot of Italian Wikipedia from 2018-04-20, 300 dimensions[65]). Since this version lacked relevant lemmas, we generated new embeddings by averaging those of semantically related words[66]. For instance, the embedding for 'covid' (which was absent in our Wikipedia snapshot) was derived by merging the embeddings of 'coronavirus', 'virus', 'pandemia' (italian word for 'pandemic'), 'contagio' ('contagion'), 'influenza' ('flu'), 'SARS', and 'malattia' ('disease'). Similarly, we built vectors for 'dreem' (the EEG device used in the study), 'actigrafo' ('actigraph'), and for the anonymization codes 'PLACE', 'NAME', and 'INSTITUTION,' which encoded sensitive information, respectively, about locations, individuals, and organizations. To approximate lexical information lost during anonymization, we selected representative words for each category, averaged their embeddings, and assigned these values to the respective categories (see Supplementary Table S10 for the specific words used).

By calculating the cosine similarity between the embeddings of all the identified lemmas, we constructed a similarity matrix that captured the entire semantic space. To derive a compact set of components (i.e., lexical domains), we applied a decomposition technique, namely Non-Negative Matrix Factorization (NNMF)[67], to obtain a lower-rank approximation of the semantic similarity matrix. We first set to zero all cells in the similarity matrix with a negative cosine value (~0.04%). Then, following the methodology proposed by Alexandrof and colleagues[68], we decomposed the similarity matrix using a cross-validation procedure to determine the optimal number of components. Specifically, for each tested component count (ranging from 1 to 70), we randomly zeroed out 10% of the similarity matrix cells, estimated the features matrix $W$ and coefficient matrix $H$ through NNMF (maximum iterations 500, alternating least squares as factorization algorithm), and computed the root mean square error (RMSE) for the left-out 10% by comparing the reconstructed and original similarity measures. Since NNMF is not exact, the procedure was repeated 50 times to obtain a robust RMSE estimate for each component count. We defined the global minimum of RMSE estimates across all the explored components ($n = 51$). Finally, for each domain, we derived a domain embedding by averaging the embeddings of the top 20 lemmas with the highest $H$ coefficients (see code repository for the list of lemmas showing the highest loading for each domain). Four raters (GBe, GH, VE, and one artificial intelligence, GPT-4o) labeled each domain by examining the meaning of the lemmas and identifying common themes or shared characteristics among the words in that domain.

To identify high-level macro categories of the domains, we measured the cosine similarity between domain embeddings and applied classical multidimensional scaling[69] to the similarity matrix, reducing the representational space to two dimensions. We then performed k-means clustering (with 1000 replications[70]) and used the silhouette criterion[71] to determine the optimal number of clusters ($n = 3$).

### Domain normalization and evaluation across verbal reports

Since our set of lexical domains was defined solely based on the semantic relatedness of the lexical items, rather than their occurrence or collocations in the reports, we then computed a weighted sum of all lemma weights for each domain and verbal report.

In detail, we extracted semantically meaningful lemmas and computed the cosine similarity between word embeddings and domain embeddings to generate a lemma-by-domain matrix. To enhance the orthogonality between domains, within each domain separately, we sparsified the lemma-by-domain vector by setting to zero all the cosine similarities below a critical threshold determined using the knee-point heuristic procedure[72]. The rationale behind this procedure was to identify the transition point in the distribution of cosine similarities where the semantic relatedness between lemmas and each domain shifts to a different regime. The procedure resulted in a lemma-by-domain matrix with approximately 98% sparsity, with 50% of the lemmas in the *main* dataset (~5000) being associated with at least a domain.

After the calculation of the lemma-by-domain matrix, the cosine similarities within each domain were normalized by dividing them by the sum of the nonzero values. Subsequently, within each lemma, data were scaled by dividing them by the sum of the nonzero values, yielding a normalized lemma-by-domain matrix. The final lemma-by-domain matrix maintained the same original sparsity, with nonzero cells containing scaled values up to 1 for lemmas that appeared exclusively within a single domain (e.g., the word *pizza* retained a value of 1 as it was associated only with the *food* domain, whereas *ciotola* -bowl- had a value of ~0.48 for *food* and ~0.52 for the *objects* domain).

To construct the report-by-domain matrix, we first extracted the lemmas from each report and summed their corresponding weights using the lemma-by-domain matrix. The resulting domain scores were then normalized by dividing them by the total number of lemmas in the report. Since the final report-by-domain matrix contained continuous values, we devised a procedure to binarize the weights, allowing for the estimation of domain frequencies and co-occurrences. Specifically, for each domain, we designed two distinct null distributions of verbal reports: one based on lemmas inherently specific to a single domain (i.e., words with loadings in the lemma-by-domain matrix only for that topic) and another using lemmas shared across multiple domains. The former was used to estimate the number of true positives (i.e., sensitivity), while the latter was used to assess the number of true negatives (i.e, specificity) during the threshold determination for binarization. In detail, the first null distribution was generated by constructing a null dataset with the same number of reports and words per report as the original *main* dataset, but ensuring that each null report contained only one domain-specific lemma. From this dataset, we derived the first null report-by-domain matrix. For the second null distribution, we followed a similar procedure but populated each null report with a set of lemmas that were not specific to a single domain, matching their occurrences in the original dataset. This allowed us to construct a second null report-by-domain matrix. Finally, we binarized the original report-by-domain matrix by determining a threshold for each domain at which sensitivity and specificity were equivalent, ensuring that the overall balanced accuracy exceeded 80%. The procedure identified 32 domains out of 51 with an average accuracy of 91 ± 4.2%.

From the binarized report-by-domain matrix, we estimated the absolute frequencies of domain occurrences across dream and wakefulness reports, along with their standard deviations across individuals (Table 2).

## Quantify the relationship between dimensions and domains

Analyses employed generalized linear mixed-effects (GLME) models that explicitly account for participant-level variance and relevant covariates, thus controlling for interindividual differences in language use and verbosity. In particular, to examine how hypothesis-driven semantic dimensions and data-driven lexical domains relate both internally and to each other, we conducted a series of analyses assessing their interdependence and convergence. We aimed to determine how semantic dimensions correlate among themselves and to what extent the two frameworks capture overlapping or distinct aspects of dream content. To quantify these associations, we applied three different GLME models. As for the dimensions, we created a GLME model (optimizer Nelder-Mead simplex algorithm) using each pairwise combination of dimensions as predictors and predicted variables. Sex, age, educational level, and the outcome of the BADA test were included as covariates of no interest, while the participant was treated as a random effect to account for inter-participant variance (see Supplementary Table S11 for a comprehensive description and naming conventions of all the variables collected). The BADA score was incorporated into all examined models as an indicator of individual verbosity. Given that longer reports are more likely to exhibit greater semantic richness, this measure is expected to correlate with higher scores across lexical domains and dimensions. While our primary focus remains on the semantic content of each report, we consider the BADA score to be an independent and reliable metric for assessing an individual's verbosity. For these models, residuals were assumed to be normally distributed conditional on the model structure, but this was not formally tested. In Wilkinson's notation, for each pairwise combination $i\,j$ of the dimensions:

$$\text{dimension}_j \sim 1 + \text{dimension}_i + \text{sex} + \text{age} + \text{education} \\ + \text{BADA} + (1|\text{Participant}) \tag{1}$$

Beta coefficients and $p$-values related to the dimensions were collected and corrected for multiple comparisons using the false discovery rate (FDR, q < 0.05, ref. [73]). Dimensions were then sorted based on their similarities and presented in Fig. 1A, upper section.

A similar procedure was applied to the lexical domains, with the only difference being that we modeled the distribution of the predicted variable as binomial:

$$\text{domain}_j \sim 1 + \text{domain}_i + \text{sex} + \text{age} + \text{education} \\ + \text{BADA} + (1|\text{Participant}) \tag{2}$$

For these models, assumptions were defined by the binomial variance structure and logit link function appropriate for binary outcomes. The domains were then sorted based on their Jaccard index, which represented the overlap between each pairwise combination of domains, and models with a significant domain coefficient (q < 0.05) presented in Fig. 1A, upper section.

Finally, to measure the association between domain and dimensions, we defined with the same approach a GLME model using each domain as the predicted variable, and each dimension as the predictor:

$$\text{domain}_j \sim 1 + \text{dimension}_i + \text{sex} + \text{age} + \text{education} \\ + \text{BADA} + (1|\text{Participant}) \tag{3}$$

Significant results for the beta coefficient of the dimension (q < 0.05) were represented in Fig. 1A, lower section.

## Validation of dimensions and domains

To evaluate the stability and reliability of the semantic dimensions and lexical domains identified in the main dataset, we conducted validation analyses using the *lockdown* dataset, which comprised 351 dream reports (reports per participant: 4.4 ± 2.8, min:1, max:14). The reports were treated and preprocessed using the same approach as described for the *main*

dataset. Regarding dimensions, dream reports were fed into the three LLMs, and ratings were collected following the same procedure outlined above. For the lexical domains, we used the 32 domains identified in the *main* dataset and applied the same procedure and parameters described above to generate the weighted report-by-domain matrix. This matrix was then binarized using the thresholds identified in the *main* dataset and domain frequencies in reports were evaluated along with their standard deviations across individuals (Table 2).

To measure the consistency among dimensions and domains in the *lockdown* dataset, we extracted beta coefficients using GLME models defined in [1] and [2]. Since the BADA score was not available in the *lockdown* dataset, we used the logarithm of the average word count across participants' reports as a proxy for verbosity. In the *main* dataset, this verbosity estimate approximated the original BADA score (Spearman's ρ = 0.168, p = 0.016). Beta coefficients for dimensions and Jaccard indices for domains are reported in Supplementary Fig. S4, along with the similarity between dimensions and domains across the two datasets.

## Differences between dream and wakefulness reports

To examine how dreaming relates to wakeful mentation, we compared the semantic structure of reports across vigilance states, identifying both overlapping and state-specific features. Accordingly, we implemented GLME models with vigilance state (dream vs. wakefulness) included as the primary regressor of interest. For each dimension or domain $j$, we defined:

$$\text{dimension}_j \sim 1 + \text{vigilance\_state} + \text{sex} + \text{age} + \text{education} \\ + \text{BADA} + (1|\text{Participant}) \tag{4}$$

$$\text{domain}_j \sim 1 + \text{vigilance\_state} + \text{sex} + \text{age} \\ + \text{education} + \text{BADA} + (1|\text{Participant}) \tag{5}$$

To enhance the robustness of the coefficient estimation associated with the vigilance state, we conducted a permutation test ($n = 5000$). Specifically, we shuffled the vigilance state binary variable within each participant, ensuring that the repeated measures structure of the data was preserved. The non-parametric $p$-values for the beta coefficient of the report type were estimated by fitting a generalized Pareto distribution to the tails of the null distribution[74]. Finally, the $p$-values for the vigilance state coefficients were corrected for multiple comparisons across all dimensions or domains using FDR (q < 0.05). The performance measures of the GLME models—including the full-model adjusted $R^2$, the full-model parametric $p$-value, coefficient estimate with the 95% confidence intervals for the vigilance state variable, Cohen's d, as well as parametric and non-parametric $p$-value estimates, along with the q-value adjusted for the latter—are reported for each dimension and domain in Table 2 and 3, respectively. The Likert points and frequencies reported in Fig. 1B were adjusted for age, sex, education level, and BADA by estimating the residuals—either continuous for dimensions or binomial for domains—using the GLME model.

## Impact of individual characteristics on dream and wakefulness reports

We next examined how individual traits influenced report content, with particular attention to factors exerting differential effects across vigilance states (i.e., interactions between each predictor and vigilance state). This approach was chosen to isolate trait-related influences that specifically modulate dreaming, rather than general effects on language production or narrative focus that could equally affect reports collected during wakefulness. By identifying traits that differentially shape dream versus waking descriptions, we aimed to highlight mechanisms uniquely associated with the phenomenology of dream experiences. To achieve this, we implemented GLME models that included all trait predictors and demographic variables simultaneously. This multivariate approach allowed us to account not only for potential confounding effects of age, sex, and education but also for

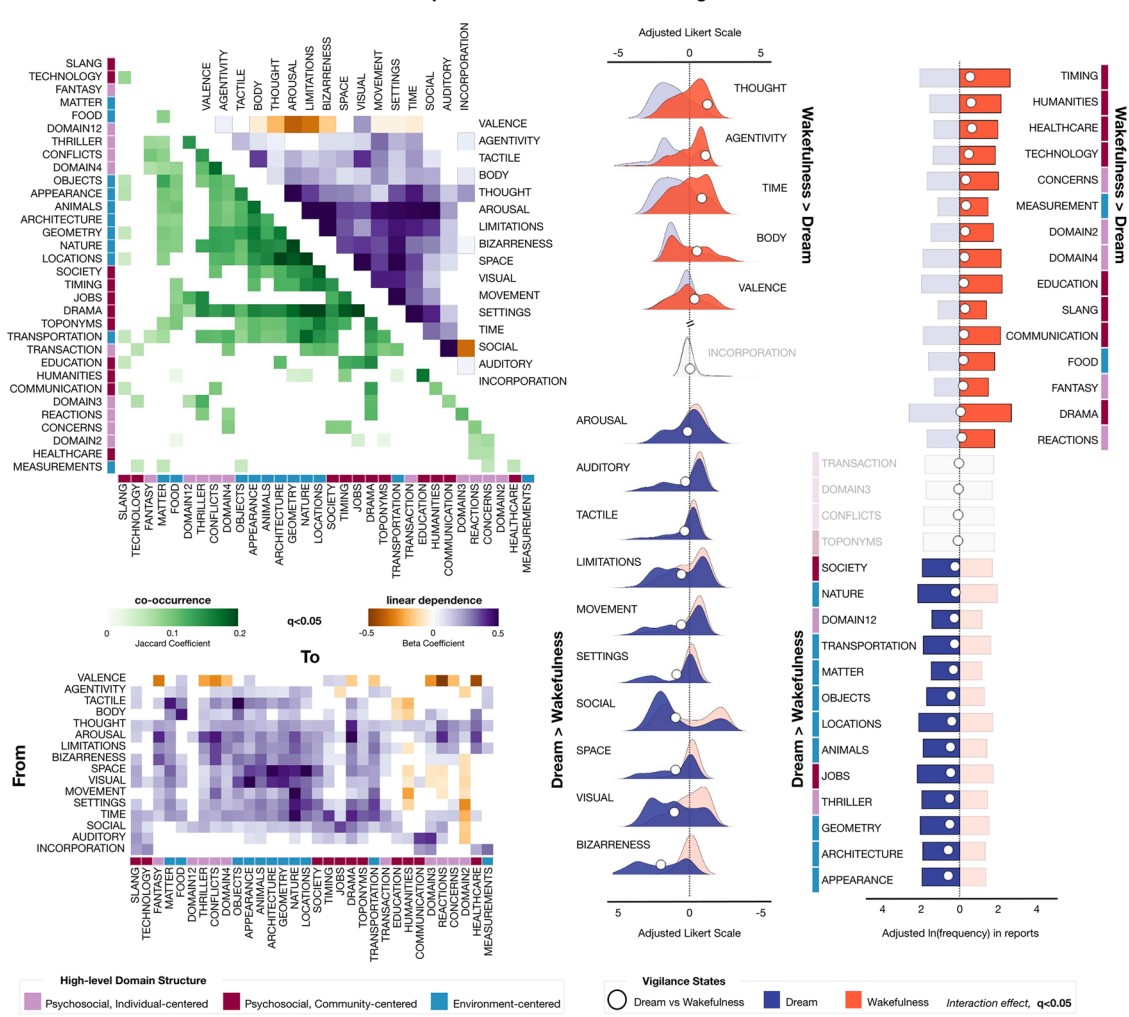

**Fig. 1 | Content analysis of dream and wakefulness reports from the main dataset.** Descriptive statistics of report content across vigilance states (i.e., wakefulness and dream), $N = 207$ participants. **A** In the upper section, matrices display associations in dream reports only (through generalized linear mixed-effect (GLME) model, using age, sex, education level, and verbosity (BADA score) as covariates of no interest; $q < 0.05$, false discovery rate (FDR) correction) among semantic dimensions and among lexical domains, and their high-level semantic organization. Matrices report the beta coefficient for dimensions and the Jaccard index to quantify the overlap of domains. In the bottom section, the matrix illustrates the associations between semantic dimensions and lexical domains (GLME model; $q < 0.05$). Empty cells indicate non-significant associations. **B** Semantic feature differences between vigilance states, i.e., wakefulness versus dream reports (GLME model; $q < 0.05$). The left section of this panel illustrates the distribution of semantic dimensions (rated on a 1-to-9 Likert scale) for wakefulness and dream reports. To facilitate result interpretation, the y-axis is interrupted, with the upper right representing features higher in wakefulness reports and the bottom left indicating those higher in dreams. The right section shows the distribution of lexical domains, quantified by their frequency (in natural logarithm) of occurrence within reports. Reported Likert points and frequencies are adjusted for age, sex, education level, and the BADA score using GLME model. White dots indicate the differences in scores or frequencies between wakefulness and dream reports. Features with a significant interaction effect ($q < 0.05$) are highlighted in red for wakefulness and blue for dreams, while non-significant features are displayed in gray.

shared variance among predictors, thereby isolating the specific contribution of each variable to dream and waking report content. The vigilance state (dream vs. wakefulness) was included as the regressor of interest, along with its interaction with all demographic and trait predictors. For a detailed description of the collected variables, refer to the Sample Characterization section, Supplementary Table S11 for naming conventions, and the MATLAB implementation provided in the code repository. In Wilkinson's notation, for each dimension or domain $j$, we defined:

$$\text{dimension}_j \sim 1 + \text{vigilance\_state} * (\text{sex} + \text{age} + \text{education} + \text{BADA}$$
$$+ \text{STAI} + \text{PSQI} + \text{ATD} + \text{MW} + \text{BSRT} + \text{ROCFr}$$
$$+ \text{MEQ} + \text{VVIQ} + \text{SCWT}) + (1|\text{Participant})$$
$$(6)$$

$$\text{domain}_j \sim 1 + \text{vigilance\_state} * (\text{sex} + \text{age} + \text{education} + \text{BADA}$$
$$+ \text{STAI} + \text{PSQI} + \text{ATD} + \text{MW} + \text{BSRT} + \text{ROCFr}$$
$$+ \text{MEQ} + \text{VVIQ} + \text{SCWT}) + (1|\text{Participant})$$
$$(7)$$

*P*-values associated with the coefficients were adjusted for multiple comparisons using FDR within each model ($q < 0.05$). The performance measures of the GLME models are reported for each dimension and domain in Supplementary Tables S2 and S3, respectively. The reported Likert points and frequencies in Fig. 2 were adjusted for demographic variables, and all questionnaires and cognitive traits included in the model, excluding the actual predictor used in the plot. Regarding the domains, we represented the interaction effect by dividing wakefulness and dream reports into two subsamples based on the median value of the predictor of interest. Then, we estimated the adjusted frequencies for the associated domain across

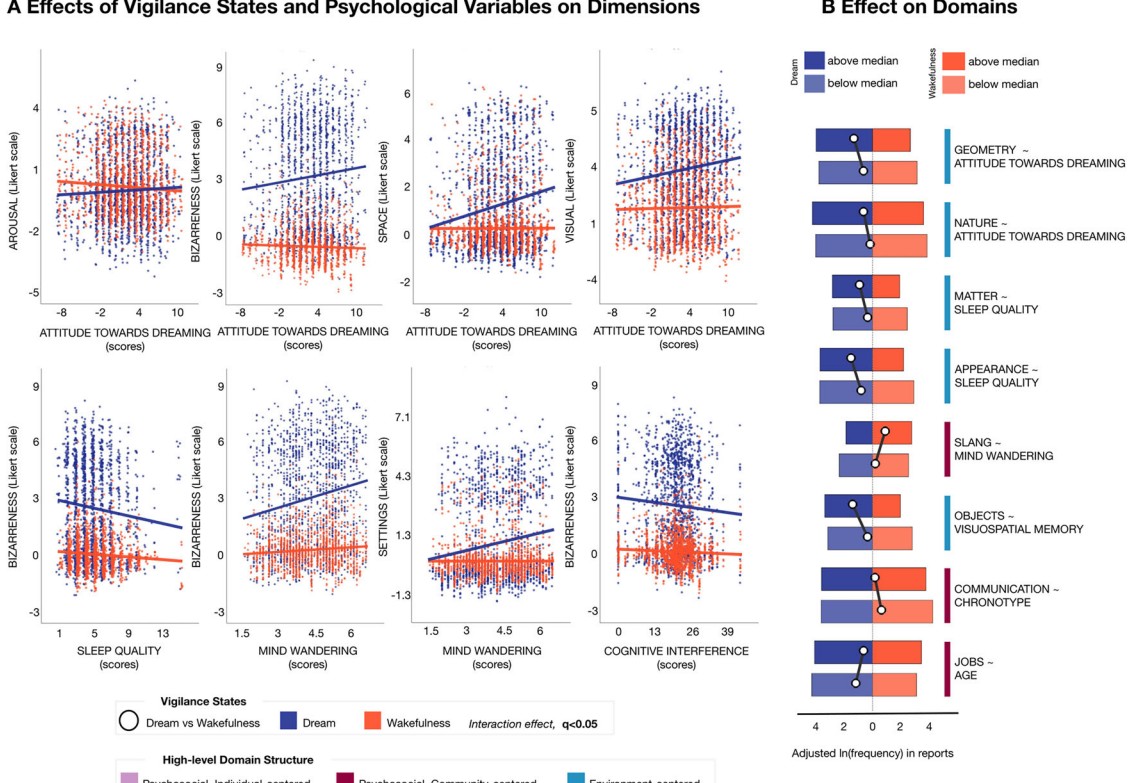

**Fig. 2 | Effect of individual variables on dream and wakefulness report content in the main dataset.** Significant effects of individual variables on semantic dimensions and domains (GLME model using psychological variables and their interaction with vigilance states as regressors of interest and age, sex, education level, and the BADA score as covariates; q < 0.05, FDR correction), N = 207 participants. The reported Likert points and frequencies are adjusted for age, sex, education level, BADA score, and all questionnaires and cognitive traits included in the model. Being sleep quality measured through the Pittsburgh Sleep Quality Index, lower scores represent better perceived sleep quality, higher scores worse perceived sleep quality. In (**A**), the scatter plots show the effect of different predictors on distinct vigilance state dimensions. Each dot represents a different report, with dream reports shown in blue and wakefulness reports displayed in red. Trend lines are also drawn for wakefulness (red) and dream (blue) reports. A slight jittering of points is applied to enhance visualization. **B** Significant effects of individual variables on lexical domains and their high-level semantic organization. White dots indicate the frequency differences between wakefulness (red) and dream (blue) reports. Bars displayed in darker and lighter colors, respectively, indicate the reports with the highest (above median) and lowest (below median) values of each predictor.

wakefulness and dream subsamples (Fig. 2B). Results for significant main effects of dimensions and domains are depicted in Supplementary Figs. S3A and S3B, respectively.

**Impact of sleep patterns on dream and wakefulness reports**
To investigate how sleep architecture influences the content of conscious experiences, we analyzed objective measures of sleep macrostructure derived from continuous actigraphic monitoring. Because actigraphy captures physiological parameters specific to nocturnal sleep, these analyses were restricted to dream reports. This approach allowed us to test whether variations in sleep architecture systematically shape the semantic and structural organization of dreams.

As detailed in previous work[75], we applied Principal Component Analysis (PCA) to reduce 24 actigraphic indices into a set of key components. The indices included, among others, *actual sleep* (or *wake*) *time* (i.e., the total time spent in sleep/wake according to the epoch-by-epoch wake/sleep categorization) and *sleep efficiency* (i.e., actual sleep time expressed as a percentage of time in bed). This analysis yielded four principal components (PCs), collectively explaining 87.74% of the variance. To aid in interpreting these PCs, we previously used mixed-effect models incorporating sleep structure measures from a subsample of participants who also wore a portable EEG system during the experimental nights. The used indices included the percentages of wakefulness, N1, N2, N3, and REM sleep, as well as the age and sex of participants. Based on the observations and the distribution of PC loadings, the four PCs were labeled as follows: sleep

fragmentation (PC1), prolonged non-N3 sleep (PC2; hereinafter referred to as "*long, light sleep*"), stable sleep with an advanced phase (PC3; "*stable advanced sleep*"), and unstable sleep with an advanced phase (PC4; "*unstable advanced sleep*"). We defined GLME models for each dimension or domain by including all predictors (demographics, questionnaires, or cognitive traits) that were found to be significant (uncorrected p-value < 0.05) in the previous analysis, as described above. Additionally, we included the report scores for the first four actigraphic PCs as regressors of interest (see Supplementary Table S11 for naming conventions):

$$\text{dimension}_j \sim 1 + (\text{significant predictors from}[6]) + PC1 + PC2 + PC3 + PC4 + (1|\text{Participant}) \quad (8)$$

$$\text{domain}_j \sim 1 + (\text{significant predictors from}[7]) + PC1 + PC2 + PC3 + PC4 + (1|\text{Participant}) \quad (9)$$

To account for potential collinearities, we included demographics, questionnaires, cognitive traits, and actigraphic data as predictors, as previous research has shown partial interactions among these variables[75]. P-values associated with the coefficients were adjusted for multiple comparisons using FDR within each model (q < 0.05). The performance measures of the GLME models are reported for each dimension and domain in Supplementary Tables S4 and S5, respectively. Results for significant PCs were represented in Supplementary Fig. S3C.

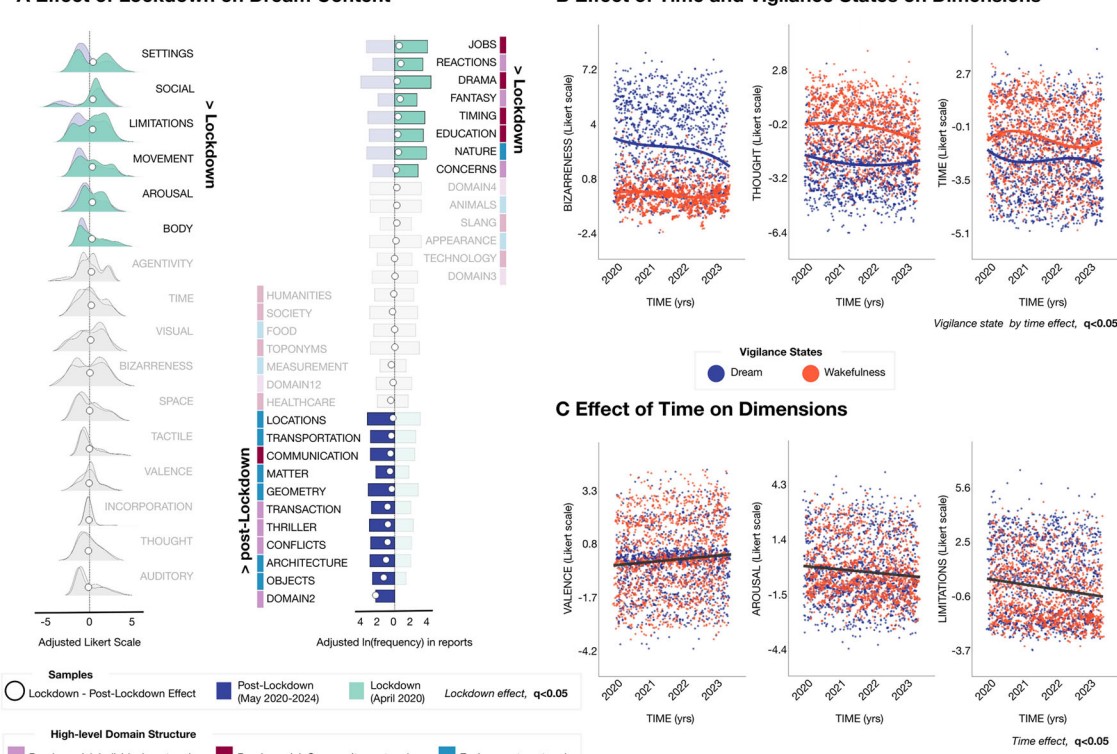

**Fig. 3 | Effect of time and external events on sleep conscious experiences.** In (**A**), comparison between dream reports collected around the time of lockdown release in Italy (lockdown dataset, from the end of April and the beginning of May 2020) and reports of the main dataset, collected from the end of May 2020 to March 2024 (GLME model, using age, sex, education level, and the BADA score as covariates; q < 0.05, FDR correction), N = 287 participants. The left column shows the results for semantic dimensions, while the right column shows the results obtained for lexical domains. White dots indicate the differences in adjusted scores or frequencies between lockdown and post-lockdown datasets. Features with a significant effect are highlighted in green for the lockdown dataset and in blue for the post-lockdown dataset, while non-significant features are displayed in gray. **B**, **C** Significant effects of the passage of time on semantic dimensions (adjusted scores for age, sex,

education level, and the BADA score as covariates using a GLME model; q < 0.05, FDR correction), N = 207 participants. The scatter plots illustrate the semantic dimensions for which we observed an interaction between the passage of time and vigilance states (bizarreness, thought, and time references; **B**) or a main effect of the passage of time (limitations, valence, arousal; **C**). Each dot represents a different report, with dream reports shown in blue and wakefulness reports displayed in red. Trend lines—obtained by identifying the optimal polynomial fit—are also drawn for wakefulness (red) and dream (blue) reports. On the x-axis, time is represented in normalized units from the beginning of data collection in March 2020 to the end of data collection in March 2024. A slight jittering of points is applied to enhance visualization.

## Impact of lockdown on dream content

To investigate the potential effects of the COVID-19 pandemic and related restriction measures on dream content, we used the *lockdown* dataset as a term of comparison with the *main* dataset. Specifically, we compared reports collected during the lockdown with a temporally matched sub-sample of the *main* dataset comprising reports obtained after May 2020, when national restrictions were lifted. Following the same approach described for models [4] and [5], we constructed a GLME model including sex, age, educational level, and the average word count per participant (WC_part) as covariates of no interest, and treating 'Participant' as a random effect. As the predictor of interest, we encoded a binary variable ('experiment') to distinguish between reports from the *main* dataset and those acquired during lockdown in the *lockdown* set. For each dimension or domain *j*, we defined:

$$\text{dimension}_j \sim 1 + \text{experiment} + \text{sex} + \text{age} + \text{education} \\ + \text{WC\_part} + (1|\text{Participant}) \tag{10}$$

$$\text{domain}_j \sim 1 + +\text{experiment} + \text{sex} + \text{age} + \text{education} \\ + \text{WC\_part} + (1|\text{Participant}) \tag{11}$$

*P*-values for the experiment coefficients were corrected for multiple comparisons across all dimensions or domains using FDR (q < 0.05). The

performance measures of the GLME models are reported for each dimension and domain in Supplementary Tables S6 and S7, respectively. The Likert points and frequencies shown in Fig. 3A were adjusted for age, sex, education level, and WC_part by estimating the residuals—either continuous for dimensions or binomial for domains—using the GLME model.

## Impact of time on dream and wakefulness reports

We next used the 4-year acquisition period of the main dataset (2020–2024) to examine long-term temporal trends in dream and waking reports. This analysis tested whether changes in dream content observed during the pandemic persisted, normalized, or evolved as the pandemic's psychological and societal impact subsided. To this end, we examined time-related variations across the post-lockdown period in both dream and waking experiences, testing for shared temporal effects and state-specific interactions to determine whether dreams retained distinctive markers of the pandemic and its aftermath. We defined GLME models similar to [4] and [5], incorporating the vigilance state as a regressor of interest, along with its interaction with time. The variable 'time' was coded by sorting all available dates in chronological order and converted into normalized ranks within the 0–1 range. For each dimension or domain *j*, we defined:

$$\text{dimension}_j \sim 1 + \text{vigilance\_state} * \text{time} + \text{sex} + \text{age} + \text{education} \\ + \text{BADA} + (1|\text{Participant}) \tag{12}$$

$$\text{domain}_j \sim 1 + \text{vigilance\_state} * \text{time} + \text{sex} + \text{age} + \text{education} \tag{13}$$
$$+ \text{BADA} + (1|\text{Participant})$$

*P*-values for the interaction between vigilance state and time were adjusted for multiple comparisons across dimensions or domains using FDR ($q < 0.05$). Similarly, the results for the main effect of time were corrected using a separate FDR procedure ($q < 0.05$). The performance measures of the GLME models are reported for each dimension and domain in Supplementary Tables S8 and S9, respectively. The effect of the passage of time on the adjusted variables was visualized in Fig. 3B, C. For illustrative purposes only, we fitted the optimal polynomial function (up to the third degree) to the adjusted data for wakefulness and dream reports independently when a significant interaction with time was detected (Fig. 3B). For main effects of time, the entire dataset was aggregated prior to trend estimation (Fig. 3C). These polynomial fits were used solely for visualization and did not contribute to the statistical modeling or inference reported in the results.

## Results

Data collection across the *main* and *lockdown* datasets yielded a combined sample of 287 adults (age range: 18–69 years; 177 female participants, 110 male participants). The *main* dataset comprised 207 participants who collectively provided 1687 dream and 1679 waking reports. In total, 2038 dream reports were analyzed across the two datasets.

To ensure both transparency and readability, the next sections focus on statistically significant results after FDR correction ($q < 0.05$), which are reported together with the corresponding effect sizes. Additional statistical information (including non-significant effects) is provided in figures/tables, and all comprehensive outputs are available in the Supplementary Information.

### Lexical domains and semantic dimensions

To characterize the semantics of dreams and their relationship with waking experiences, we applied two complementary approaches. The first was hypothesis-driven, quantifying 16 pre-defined semantic dimensions to capture overarching narrative features (Supplementary Table S1). The evaluation of each report was performed through three LLMs, whose performance was validated against external human raters and the ratings provided by an independent sample of participants who scored their own dreams (see for more details Supplementary Text and Supplementary Figs. S1 and S2). Agreement between AI-generated scores and external human ratings was consistently high across all 16 semantic dimensions (all $\rho > 0.60$; range: $\rho = 0.627$–0.999), reaching levels comparable to those observed among independent human raters (see Supplementary Fig. S1). In addition, comparisons with dreamers' self-ratings yielded a mean correlation of $\rho = 0.65$ (range: $\rho = 0.319$–0.847; see Supplementary Fig. S2), indicating substantial agreement between the AI evaluations and participants' own assessments.

The second approach was data-driven, analyzing specific lexical domains by grouping semantically related words to map the detailed thematic content of each report. We identified 32 low-level lexical domains which further clustered into three high-level categories: (i) *environment*-related domains encompass the visual, geometric, and structural properties of natural and artificial spaces, including colors, objects, spatial locations, and descriptions of buildings and nature; (ii) *community*-related domains reflect societal structures, institutions, and dynamics, incorporating terms related to education, healthcare, and culture; (iii) *individual*-related domains capture mental states, emotions, and personal reactions, referencing evaluation processes, fear, and imagination.

To assess the reliability and reproducibility of the semantic dimensions and lexical domains identified in the *main* dataset, we validated them using data from the *lockdown* dataset (Supplementary Fig. S4). Both dimensions and domains showed strong cross-dataset consistency (Spearman's $\rho = 0.87$

and $\rho = 0.55$, respectively; $p < 0.00001$), confirming the robustness of the identified semantic structures.

### Relationship between lexical domains and semantic dimensions

We first examined how the hypothesis-driven semantic dimensions and data-driven lexical domains contributed to the characterization of verbal reports and how they related to one another. These analyses aimed to assess the interdependence and convergence between the two frameworks, determining the extent to which they captured overlapping versus distinct aspects of dream content.

In the description of dream experiences (Fig. 1A, lower panel), dimensions reflecting perceptual (i.e., *tactile* and *visual*) and spatial references (*space*) were positively associated with *environment*-related domains (e.g., *space* and *locations* domain, β = 0.50382; CI = [0.44549, 0.56215]; q < 0.0001). Interestingly, *individual*-related domains exhibited a negative relationship with emotional *valence* (e.g., *conflicts* domain, β = −0.28348; CI = [−0.37889, −0.18806]; q < 0.0001), indicating that dreams with stronger internal focus tended to be more negative and with a higher *arousal* (β = 0.42411; CI = [0.35016, 0.49805]; q < 0.0001). For instance, more distressing and emotionally charged dreams clustered within the *thriller* domain, encompassing themes of danger and life-threatening situations (positively associated with *arousal* (β = 0.35814; CI = [0.28949, 0.42678]; q < 0.0001) and negatively with *valence* (β = −0.24021; CI = [−0.3291, −0.15133]; q < 0.0001). Further, *auditory* and *communication* references were particularly linked to *community*-related domains (e.g., *society* domain was positively associated to auditory experiences, β = 0.11766; CI = [0.059898, 0.17542]; q < 0.0001), suggesting a connection between verbal interactions and social elements within the reports. Additionally, narratives characterized by heightened *arousal* (β = 0.48985; CI = [0.41846, 0.56124]; q < 0.0001), frequent *setting* changes (β = 0.31289; CI = [0.25602, 0.36976]; q < 0.0001) and increased *time* references (β = 0.40523; CI = [0.34379, 0.46666]; q < 0.0001) tended to fall within a *drama* domain—an intricate blend of suspense, horror, romance, and tragedy, enriched by storytelling-related terms. Hence, the hierarchical structure of lexical domains aligned closely with the distribution of semantic dimensions, highlighting the intertwining, yet complementary, nature of these features. Each linguistic form, that is the words with their conventionally associated meanings, mapped onto broader semantic aspects, collectively shaping the narratives[59].

### Content analysis of dream and wakefulness reports

Next, we investigated how the semantic structure of experiences differs between sleep and wakefulness. This analysis aimed to identify both common features reflecting the continuity of cognition across vigilance states and distinctive characteristics specific to dreaming (Fig. 1B).

Dream reports exhibited significantly more references to perceptual (e.g., the *visual* dimension with a dream minus wakefulness Cohen's d of 1.52; q < 0.05; Table 1) and spatial features (e.g., *space* d = 1.40, *settings* d = 1.29), whereas wakefulness reports were enriched with descriptions of thoughts and metacognitive processes (*thought* d = −1.77). Emotionally, dreams were characterized by higher *arousal* (d = 0.21) and more negative *valence* (d = −0.49) compared to wakefulness experiences. Dreams also contained significantly more references to *social* interactions (d = 1.40), exhibited a marked increase in *bizarreness* (d = 2.85), and featured more references to *limitations* (d = 0.82) on the character's freedom. In contrast, wakefulness reports reflected greater *agentivity* (d = −1.58), with participants portraying themselves as more in control of their actions, aware of the *time* flow (d = −1.20), and attuned to their *body*'s needs (d = −0.73). Notably, the semantic dimension capturing references to the experimental procedure did not differ between dream and wakefulness reports (*incorporation* d = −0.05, 95th confidence intervals: −0.11 0.01, with an average ± standard deviation across all the reports of 1.2 ± 0.87 Likert points). This indicates that the study design did not introduce systematic biases in the sampling of dream and wakefulness experiences.

Dream reports exhibited a significantly higher prevalence of elements related to the *environment* lexical domain, encompassing both animate (e.g.,

**Table 1 | Performance measures of the GLME model for the prediction of semantic dimensions across vigilance states in the main dataset**

| Dimension | Full Model adj R² | p-value Full model | Report-type coefficient | Cohen's d | p-value non-parametric coefficient | q-value coefficient |
|---|---|---|---|---|---|---|
| AGENTIVITY | 0.183 | <0.00001 | −1.58, CI: −1.70 −1.46 | −1.31 | <0.00001 | **<0.00001** |
| AROUSAL | 0.096 | 0.00002 | 0.21, CI: 0.10 0.31 | 0.19 | 0.01140 | **0.01216** |
| AUDITORY | 0.044 | <0.00001 | 0.42, CI: 0.30 0.55 | 0.29 | <0.00001 | **<0.00001** |
| BIZARRENESS | 0.426 | <0.00001 | 2.85, CI: 2.73 2.98 | 1.88 | <0.00001 | **<0.00001** |
| BODY | 0.076 | <0.00001 | −0.73, CI: −0.85 −0.61 | −0.58 | <0.00001 | **<0.00001** |
| INCORPORATION | 0.014 | 0.11368 | −0.05, CI: −0.11 0.01 | −0.08 | 0.08945 | 0.08945 |
| LIMITATIONS | 0.106 | <0.00001 | 0.82, CI: 0.69 0.95 | 0.54 | <0.00001 | **<0.00001** |
| THOUGHT | 0.346 | <0.00001 | −1.77, CI: −1.88 −1.66 | −1.20 | <0.00001 | **<0.00001** |
| MOVEMENTS | 0.076 | <0.00001 | 0.83, CI: 0.70 0.96 | 0.56 | <0.00001 | **<0.00001** |
| SETTINGS | 0.177 | <0.00001 | 1.29, CI: 1.18 1.40 | 0.99 | <0.00001 | **<0.00001** |
| SOCIAL | 0.129 | <0.00001 | 1.40, CI: 1.22 1.58 | 0.69 | <0.00001 | **<0.00001** |
| SPACE | 0.211 | <0.00001 | 1.40, CI: 1.30 1.51 | 1.20 | <0.00001 | **<0.00001** |
| TACTILE | 0.064 | <0.00001 | 0.49, CI: 0.40 0.58 | 0.60 | <0.00001 | **<0.00001** |
| TIME | 0.173 | <0.00001 | −1.20, CI: −1.33 −1.08 | −0.72 | <0.00001 | **<0.00001** |
| VALENCE | 0.063 | <0.00001 | −0.49, CI: −0.59 −0.38 | −0.43 | <0.00001 | **<0.00001** |
| VISUAL | 0.190 | <0.00001 | 1.52, CI: 1.38 1.66 | 1.08 | <0.00001 | **<0.00001** |

The vigilance state (report-type coefficient) was used as a regressor of interest; age, sex, education level, and the BADA score were used as covariates of no interest; q < 0.05, false discovery rate (FDR) correction. Dimension labels in bold are those surviving FDR correction. Cohen's d refers to the effect of the report-type coefficient.

*animals*, d = 0.93, present in 18.4% of dream reports; q < 0.05; see Tables 2 and 3) and inanimate entities (e.g., *nature* d = 0.48, 29.8%, *locations* d = 0.84, 27.6%, *objects* d = 0.40, 13.5%, and their physical *appearance* d = 1.13, 19.9%). References to *measurement* and *food* did not follow this trend. In contrast, wakefulness reports were dominated by elements from the *individual* and *community* lexical categories. Exceptions included terms associated with broad descriptive properties (*domain12*, 10.1%), *thriller*-like themes (21.9%), societal structures (*society*, 20.4%), and occupational references (*jobs*, 32.7%), which were more frequently observed in dreams.

### Effect of individual variables on dream and wakefulness report content

We next examined how individual differences influenced the content of reports, focusing on factors that had distinct effects on wakefulness and dream experiences (i.e., interaction between each predictor and the vigilance state, q < 0.05; Fig. 2A; Supplementary Tables S2 and S3). This approach allowed us to identify trait-related influences that specifically characterize dreaming, beyond general effects on language or narrative style common to both vigilance states.

Attitude toward dreaming emerged as a key predictor, selectively enhancing several semantic dimensions in dream narratives but not in wakefulness, including emotional *arousal* (interaction β = 0.05; CI = [0.02, 0.08], q = 0.04492), *bizarreness* (β = 0.07; CI = [0.03, 0.10]; q = 0.00296), spatial features (*space*, β = 0.07; CI = [0.04, 0.10]; q = 0.00017), and *visual* perception (β = 0.07; CI = [0.03, 0.10]; q = 0.02136). Additionally, this trait predicted references to geometric patterns (*geometry*, β = 0.09; CI = [0.04, 0.15]; q = 0.01627) and navigation in natural environments (*nature*, β = 0.07; CI = [0.03, 0.12]; q = 0.04748) to a different degree across dream and wakefulness experiences. Notably, subjective sleep quality (β = −0.09; CI = [−0.15, −0.03]; q = 0.02496), lower vulnerability to cognitive interference (β = −0.03; CI = [−0.05, −0.01]; q = 0.04665), and a greater propensity for mind-wandering (β = 0.27; CI = [0.16, 0.38]; q = 0.00005) each contributed to heightened dream *bizarreness*. In particular, proneness to mind-wandering was also associated with more frequent shifts in dream *settings* (β = 0.23; CI = [0.14, 0.33]; q = 0.00008), which may increase the subjective perception of bizarreness. Lower perceived sleep quality was associated with references to *matter*- (β = 0.22; CI = [0.08, 0.36];

q = 0.03306) and *appearance*-related (β = 0.17; CI = [0.06, 0.29]; q = 0.04854) domains (Fig. 2B). Moreover, individuals with higher visuo-spatial memory abilities reported dreams with an increased frequency of *object* references (β = 0.09; CI = [0.04, 0.14]; q = 0.00564). Younger participants reported *job*-related details more frequently in dreams compared to wakefulness reports (β = −0.03; CI = [−0.04, −0.01], q = 0.02973). Additionally, an evening chronotype was linked to a greater focus on *communication*-related content in wakefulness reports compared to dreams (β = 0.03; CI = [0.01, 0.04]; q = 0.04769).

Further analyses examined the relationship between objective sleep patterns, as derived by continuous actigraphic monitoring, and dream content (Supplementary Fig. S3, Supplementary Tables S4 and S5). Among the examined indices, those associated with long, light sleep showed a significant positive association with the frequency of *setting* shifts in dreams (β = 0.09, CI = [0.03, 0.16]; q = 0.02325).

### Effect of time and external events on sleep conscious experiences

Finally, we investigated how external events and the passage of time influenced dream content. To this aim, we performed two complementary analyses. While these analyses cannot establish causal relationships, they allowed us to identify potential associations between large-scale societal events, time-dependent factors, and the phenomenology of sleep-related conscious experiences.

First, we compared reports acquired during the Italian COVID-19 lockdown with a temporally matched subsample of the main dataset collected after the easing of restrictions (from the end of May 2020 onward), ensuring no temporal overlap between the two (q < 0.05; Fig. 3A; Supplementary Tables S6–S7). Throughout the period of restrictions, dreams contained significantly more references to *limitations* (d = 0.46; CI = [0.15, 0.71]; q = 0.01496), *social* interactions (d = 0.48; CI = [0.12, 0.82]; q = 0.02800), *settings* (d = 0.41; CI = [0.22, 0.78]; q = 0.00482), *body* (d = 0.36; CI = [0.09, 0.60]; q = 0.02800), and emotional *arousal* (d = 0.50; CI = [0.17, 0.63]; q = 0.00482). Additionally, there was an increase in references to fantastical (*fantasy*, d = 0.50; CI = [0.16, 0.95]; q = 0.00984) or *drama* (d = 1.09; CI = [0.35, 1.01]; q = 0.00020) elements, work-related themes (*jobs*, d = 0.88; CI = [0.58, 1.15]; q < 0.00001), and temporal references

**Table 2 | Absolute frequencies of lexical domains**

| High-level Domain | Domain | Main dataset, dream reports | Main dataset, wakefulness reports | Lockdown dataset, dream reports |
|---|---|---|---|---|
| Community-centered | Toponyms | 19.03 ± 32.69 % | 16.05 ± 25.00 % | 19.52 ± 25.49 % |
| Community-centered | Healthcare | 8.53 ± 14.05 % | 21.75 ± 25.48 % | 6.07 ± 14.42 % |
| Community-centered | Jobs | 32.66 ± 27.40 % | 16.11 ± 25.00 % | 54.33 ± 33.47 % |
| Community-centered | Drama | 60.53 ± 32.00 % | 65.39 ± 42.18 % | 85.38 ± 21.16 % |
| Community-centered | Society | 20.42 ± 33.33 % | 13.89 ± 23.08 % | 14.78 ± 20.58 % |
| Community-centered | Humanities | 11.30 ± 19.69 % | 28.64 ± 34.95 % | 14.29 ± 25.68 % |
| Community-centered | Timing | 26.63 ± 24.72 % | 60.60 ± 40.95 % | 38.73 ± 31.47 % |
| Community-centered | Technology | 7.93 ± 12.50 % | 17.67 ± 25.00 % | 11.54 ± 22.16 % |
| Community-centered | Education | 22.29 ± 24.81 % | 31.76 ± 34.88 % | 34.66 ± 28.68 % |
| Community-centered | Communication | 19.22 ± 29.64 % | 26.10 ± 28.46 % | 9.90 ± 15.80 % |
| Community-centered | Slang | 5.56 ± 10.00 % | 8.06 ± 12.15 % | 7.44 ± 18.08 % |
| Environment-centered | Food | 12.14 ± 18.18 % | 16.86 ± 28.25 % | 11.04 ± 18.78 % |
| Environment-centered | Objects | 13.53 ± 22.22 % | 6.60 ± 11.11 % | 3.49 ± 9.89 % |
| Environment-centered | Matter | 9.67 ± 16.67 % | 5.62 ± 9.09 % | 4.94 ± 12.13 % |
| Environment-centered | Locations | 27.58 ± 28.89 % | 14.02 ± 22.22 % | 18.74 ± 24.41 % |
| Environment-centered | Geometry | 24.25 ± 28.46 % | 10.06 ± 16.67 % | 16.16 ± 19.50 % |
| Environment-centered | Nature | 29.80 ± 28.27 % | 21.13 ± 33.33 % | 47.86 ± 33.91 % |
| Environment-centered | Measurements | 6.35 ± 11.11 % | 10.26 ± 16.67 % | 3.44 ± 9.90 % |
| Environment-centered | Architecture | 20.16 ± 30.77 % | 7.34 ± 12.50 % | 6.02 ± 13.14 % |
| Environment-centered | Animals | 18.44 ± 32.69 % | 8.74 ± 12.50 % | 20.63 ± 25.36 % |
| Environment-centered | Appearance | 19.94 ± 30.00 % | 7.83 ± 12.15 % | 24.59 ± 27.83 % |
| Environment-centered | Transportation | 19.14 ± 30.58 % | 12.04 ± 20.00 % | 13.27 ± 22.96 % |
| Individual-centered | DOMAIN2 | 10.13 ± 16.67 % | 16.21 ± 25.00 % | 0.14 ± 1.24 % |
| Individual-centered | DOMAIN3 | 15.49 ± 25.00 % | 13.65 ± 22.22 % | 18.27 ± 24.48 % |
| Individual-centered | DOMAIN4 | 18.79 ± 30.77 % | 28.17 ± 25.71 % | 30.73 ± 28.83 % |
| Individual-centered | Reactions | 14.16 ± 22.22 % | 17.34 ± 29.64 % | 32.71 ± 30.28 % |
| Individual-centered | DOMAIN12 | 9.83 ± 16.35 % | 6.59 ± 11.11 % | 9.39 ± 21.73 % |
| Individual-centered | Thriller | 21.89 ± 33.33 % | 9.68 ± 16.67 % | 10.61 ± 21.55 % |
| Individual-centered | Fantasy | 7.50 ± 14.29 % | 10.18 ± 16.67 % | 15.02 ± 22.91 % |
| Individual-centered | Concerns | 14.97 ± 22.22 % | 23.83 ± 33.33 % | 18.04 ± 24.04 % |
| Individual-centered | Conflicts | 18.30 ± 30.00 % | 15.88 ± 25.00 % | 6.76 ± 12.89 % |
| Individual-centered | Transaction | 16.61 ± 25.00 % | 14.76 ± 22.86 % | 5.81 ± 11.19 % |

Absolute frequencies of domain occurrences in dream and wakefulness reports, with standard deviations across individuals in the main Dataset, and absolute frequencies in the lockdown Dataset (q < 0.05). In the first column, high-level semantic categories. In the second column, the lexical domain labels, as assigned by the raters.

(*timing*, d = 0.51; CI = [0.25, 0.82]; q = 0.00071). Importantly, the correlation structure among semantic dimensions and lexical domains was highly consistent across the *main* and *lockdown* datasets, indicating that the observed differences reflect a genuine shift in content rather than diachronic changes in language (Supplementary Fig. S4).

Second, we examined longitudinal changes within the *main* dataset (2020–2024) to evaluate how dream and waking report features evolved from the pandemic period onward (q < 0.05; Fig. 3B; Supplementary Tables S8 and S9). Over time, dream *bizarreness* decreased (interaction β = −0.74; CI = [−1.17, −0.31]; q = 0.00956), possibly reflecting a

**Table 3 | Performance measures of the GLME model for the prediction of lexical domains across vigilance states in the main dataset**

| | Domain | Full Model adj $R^2$ | p-value Full model | report-type coefficient | Cohen's d | p-value non-parametric coefficient | q-value coefficient |
|---|---|---|---|---|---|---|---|
| | **DOMAIN2** | 0.066 | < 0.00001 | -0.53, CI: -0.75 -0.32 | -0.27 | 0.00005 | **0.00008** |
| | DOMAIN3 | 0.010 | 0.00339 | 0.11, CI: -0.08 0.30 | 0.09 | 0.30674 | 0.31663 |
| | **DOMAIN4** | 0.023 | < 0.00001 | -0.53, CI: -0.69 -0.37 | -0.37 | < 0.00001 | **< 0.00001** |
| | **Food** | 0.005 | 0.00047 | -0.37, CI: -0.56 -0.18 | -0.22 | 0.00129 | **0.00165** |
| | **Objects** | 0.047 | < 0.00001 | 0.80, CI: 0.57 1.02 | 0.40 | < 0.00001 | **< 0.00001** |
| | **Reactions** | 0.015 | 0.37539 | -0.21, CI: -0.39 -0.02 | -0.16 | 0.03753 | **0.04289** |
| | **DOMAIN12** | 0.015 | < 0.00001 | 0.50, CI: 0.24 0.75 | 0.19 | 0.00006 | **0.00008** |
| | **Matter** | 0.007 | 0.00007 | 0.56, CI: 0.31 0.82 | 0.31 | < 0.00001 | **< 0.00001** |
| | Toponyms | 0.034 | 0.00966 | 0.18, CI: 0.00 0.36 | 0.12 | 0.13689 | 0.15106 |
| | **Thriller** | 0.038 | < 0.00001 | 0.99, CI: 0.79 1.19 | 0.54 | < 0.00001 | **< 0.00001** |
| | **Healthcare** | 0.054 | < 0.00001 | -1.17, CI: -1.38 -0.97 | -0.64 | < 0.00001 | **< 0.00001** |
| | **Locations** | 0.061 | < 0.00001 | 0.84, CI: 0.67 1.02 | 0.53 | < 0.00001 | **< 0.00001** |
| | **Geometry** | 0.051 | < 0.00001 | 1.00, CI: 0.81 1.19 | 0.67 | < 0.00001 | **< 0.00001** |
| | **Nature** | 0.034 | < 0.00001 | 0.48, CI: 0.32 0.64 | 0.34 | < 0.00001 | **< 0.00001** |
| | **Fantasy** | 0.001 | 0.08989 | -0.32, CI: -0.55 -0.08 | -0.15 | 0.01126 | **0.01335** |
| | **Measurements** | 0.007 | 0.00003 | -0.56, CI: -0.82 -0.30 | -0.22 | < 0.00001 | **< 0.00001** |
| | **Architecture** | 0.032 | < 0.00001 | 1.08, CI: 0.86 1.29 | 0.61 | < 0.00001 | **< 0.00001** |
| | **Animals** | 0.071 | < 0.00001 | 0.93, CI: 0.72 1.14 | 0.50 | < 0.00001 | **< 0.00001** |
| | **Concerns** | 0.035 | < 0.00001 | -0.62, CI: -0.79 -0.44 | -0.34 | < 0.00001 | **< 0.00001** |
| | **Appearance** | 0.074 | < 0.00001 | 1.13, CI: 0.92 1.34 | 0.54 | < 0.00001 | **< 0.00001** |
| | **Jobs** | 0.046 | < 0.00001 | 0.97, CI: 0.80 1.14 | 0.60 | < 0.00001 | **< 0.00001** |
| | Conflicts | 0.013 | 0.47031 | 0.16, CI: -0.02 0.34 | 0.11 | 0.15134 | 0.16143 |
| | **Drama** | 0.072 | < 0.00001 | -0.25, CI: -0.40 -0.11 | -0.14 | 0.01096 | **0.01335** |
| | **Society** | 0.027 | < 0.00001 | 0.48, CI: 0.30 0.66 | 0.28 | < 0.00001 | **< 0.00001** |
| | **Humanities** | 0.098 | < 0.00001 | -1.20, CI: -1.39 -1.02 | -0.69 | < 0.00001 | **< 0.00001** |
| | **Transportation** | 0.010 | < 0.00001 | 0.51, CI: 0.32 0.70 | 0.33 | < 0.00001 | **< 0.00001** |
| | **Timing** | 0.167 | < 0.00001 | -1.54, CI: -1.69 -1.39 | -1.03 | < 0.00001 | **< 0.00001** |
| | **Technology** | 0.022 | < 0.00001 | -0.86, CI: -1.07 -0.65 | -0.48 | < 0.00001 | **< 0.00001** |
| | **Education** | 0.047 | < 0.00001 | -0.47, CI: -0.63 -0.32 | -0.33 | < 0.00001 | **< 0.00001** |
| | Transaction | 0.003 | 0.77129 | 0.07, CI: -0.11 0.25 | 0.09 | 0.45791 | 0.45791 |
| | **Communication** | 0.032 | < 0.00001 | -0.46, CI: -0.62 -0.30 | -0.25 | < 0.00001 | **< 0.00001** |
| | **Slang** | 0.015 | < 0.00001 | -0.46, CI: -0.73 -0.20 | -0.16 | 0.00108 | **0.00144** |

The vigilance state (report-type coefficient) was used as a regressor of interest; age, sex, education level, and the BADA score were used as covariates of no interest; q < 0.05, false discovery rate (FDR) correction. Domain labels in bold are those surviving FDR correction. Cohen's d refers to the effect of the report-type coefficient.

normalization of dream content as pandemic-related stressors subsided. References to *thoughts* (β = 0.63; CI = [0.24, 1.02]; q = 0.00956) and *time* (β = 0.69; CI = [0.26, 1.12]; q = 0.00956) also declined, particularly in wakefulness reports, suggesting a shift in cognitive focus during waking life. Furthermore, across both dream and wakefulness reports (main effect of time, q < 0.05; Fig. 3C), emotional *valence* tended to increase over time (β = 0.59; CI = [0.27, 0.91]; q = 0.00079), while *arousal* (β = −0.79; CI = [−1.17, −0.41]; q = 0.00043) and references to *limitations* (β = −1.09; CI = [−1.52, −0.65]; q = 0.00001) and to the *society* (β = −0.95; CI = [−1.43, −0.46]; q = 0.00384) domain progressively diminished.

## Discussion

The generative potential of the dreaming process, its capacity to spontaneously weave vivid, coherent narratives, finds a striking echo in Mary Shelley's account of the oneiric experience that inspired the Frankenstein novel: "*When I placed my head on my pillow, […] my imagination, unbidden, possessed and guided me, gifting the successive images that arose in my mind with a vividness far beyond the usual bounds of reverie*"[76]. Her words capture the essence of the dreaming mind as both deeply personal and inherently creative.

From being the ineffable product of human consciousness, over the past 125 years, dreams have become an object of rigorous scientific observation. The study of how oneiric experiences are shaped by the self and personal experience has progressed from clinical and interpretive approaches focused on individual cases to structured, population-level analyses, where trained raters assess themes, emotions, and characters embedded in dream narratives. These approaches have provided fundamental insights into the nature of dream experiences, highlighting their complexity and connection to waking life. In this study, we leveraged state-of-the-art NLP techniques to analyze a large collection of dream reports, achieving a level of semantic characterization that would be prohibitively time-consuming and less consistent using traditional human scoring. Through this systematic approach, we demonstrated that dream content is shaped not only by each individual's unique characteristics but also by broader, generalizable traits and shared external events. This evidence points toward the coexistence of mechanisms that are broadly expressed across individuals and others that depend on each person's unique profile.

Our findings indicate that dream reports contain a constellation of features that, when contrasted with waking reports, more closely resemble patterns characteristic of narrative or cinematic description[60,77]. Like a movie unfolding on the screen, dreams tend to follow dynamic scene-by-scene structures described at the awakening as vivid, bizarre and immersive (Fig. S5). Perceptual details -particularly visuo-spatial elements- dominate the experience, with the dreamer more frequently adopting a spectator-like role, showing a heightened tendency to observe rather than actively engaging in the unfolding events. This shift in viewpoint, combined with frequent setting changes and abrupt conceptual transitions, contributes to the fragmented and discontinuous nature of dream narratives, enhancing their bizarreness relative to wakefulness. While waking experiences are typically rooted in daily life dynamics and constrained by the individual's social interactions, concerns, and goals, dreams are projected into more fluid and immersive environments. In contrast to wakefulness—which maintains an egocentric frame of reference anchored to the individual's physical position and temporal context—dreaming consciousness generates allocentric perceptual simulations. This representational shift effectively decenters the experiential locus, embedding the dreamer within rather than positioning them relative to the simulated environment. The results are consistent with the idea that elements of waking life might be transformed during sleep, with fragments of reality reshaped and reorganized into novel oneiric narratives. Rather than constituting a direct replay of daily experiences, dreams may offer a *hyper-associative* reinterpretation of past events and future expectations, weaving together apparently distant elements into coherent, though often bizarre, scenarios[11].

The distinctive narrative and perceptual organization revealed in this study supports theoretical models that conceive of dreaming as an immersive virtual simulation of waking life. Within *Threat Simulation Theory*[78], dreams are viewed as an evolutionary mechanism for rehearsing adaptive responses to emotionally charged or threatening situations, whereas the later *Social Simulation Theory*[79] emphasizes their role in maintaining and refining social skills through simulated interactions. The prevalence of emotionally intense and socially rich scenarios observed in our data accords with these frameworks, suggesting that dreams may selectively amplify salient aspects of waking experience. Importantly, our observation that dreams exhibit a greater proportion of negative emotions and higher emotional intensity than wake reports also aligns with contemporary accounts that emphasize the role of dreaming in the processing of emotional

memories and affective regulation[80]. At the same time, the immersive, self-contained nature of dream narratives resonates with Freud's classical view of dreams as the "guardians of sleep"[81], transforming internal tensions and external stimuli into coherent, internally generated imagery that helps preserve the continuity of rest. In this context, the vivid and absorbing phenomenology of dreaming may itself serve a protective function, engaging the mind in internally generated simulations that reduce sensitivity to external sensory input and sustain the stability of the sleeping state[82].

Beyond quantifying the phenomenological representation of dreaming, our study revealed how individual trait and state variables shape the content of conscious experiences, exerting distinct effects across vigilance states. While each individual perceives, interprets, and remembers their experiences—whether in wakefulness or in sleep—in a unique way, generalizable traits consistently appear to influence the semantics of dreams. Thus, demographic variables are almost exclusively associated with the features of verbal reports independently of the vigilance state they describe (see Supplementary Tables S2 and S3), whereas a higher interest in dreams and their significance aids more engaging, immersive experiences during sleep, but not in wakefulness. Interestingly, a positive *attitude towards dreaming* has also been associated with a higher probability of waking up in the morning with at least the perception of having been dreaming just a few seconds before[75]. Two main non mutually exclusive hypotheses have been proposed to explain the relationship between interest in dreams and dream recall that may be also relevant to explain our present findings[40,41]. In fact, individuals with a stronger interest in dreaming may present a greater, pervasive focus on their inner experiences and may thus be more likely to remember and report the vivid details of their dreams. It is unclear, though, why this heightened focus would affect only the reporting of specific perceptual aspects of the experience. An alternative interpretation posits that a heightened interest for dreaming might be the surface of a deeper mechanism, yet to be explored, allowing individuals to experience more vivid, immersive dreams. This enhanced dream phenomenology could, in turn, facilitate dream recall and reinforce the individuals' interest towards dreaming.

Moreover, our findings indicate that dream bizarreness is associated with a higher tendency of the individuals to mind-wander, which also drives frequent shifts in narrative settings. This is in line with accounts suggesting that dreaming and mind-wandering may share a common neural and cognitive foundation[27,83]. Individuals who frequently mind-wander may have an enhanced propensity to engage in spontaneous, self-generated experiences, independently of external stimuli. Following this reasoning, dreams may represent an intensified form of mind-wandering occurring during sleep. Consistently, we showed in a recent study that proneness to mind-wander is associated with a higher dream recall, potentially reflecting a heightened dream generation process[75]. While the precise physiological mechanism underlying this increased tendency to engage in internally driven thoughts is yet to be fully understood, some evidence points to a possible role of the so-called default mode network (DMN), a set of brain areas associated with self-reflection and inward thinking[84]. Our results provide further support to the continuity between waking- and sleep-related mentation demonstrating that a stronger tendency to disengage from external stimuli and focus on the spontaneous flow of internally generated thoughts contributes to shaping the fragmented and discontinuous nature of dreams.

Surprisingly, overall sleep patterns—whether measured objectively through actigraphy or subjectively reported—had a relatively weak association with dream content. However, specific aspects of sleep did influence particular dream characteristics. Actigraphy-based measures of sleep macrostructure revealed that prolonged light sleep was associated with more frequent shifts in dream settings. Similarly, self-reported sleep quality was related to the bizarreness of dream reports, further supporting the idea that setting shifts contribute to dream bizarreness. These findings align with previous research suggesting that dreams become progressively richer across the night[4,85,86]. Additionally, physiologically 'lighter' sleep stages (N1-N2) have consistently been associated with more complex dream content compared to 'deeper' stages (N3)[87,88]. Therefore, longer sleep duration may

increase the likelihood of collecting reports of more elaborate and complex dreams. Notably, lower self-reported sleep quality also appeared to influence waking reports more than dream reports in certain aspects. Specifically, individuals who rated their sleep as poorer tended to include fewer references to the appearance and content of their surrounding environment. This observation is in line with evidence indicating that sleep deprivation and poor sleep quality can lead to social withdrawal and a diminished interest in external interactions[89].

Our study finally shows how external emotionally salient events, in this case the COVID-19 pandemic, might affect dream experiences and how such effects develop over long time spans. Dreams are thought to play a crucial role in learning, memory consolidation, and emotion regulation. According to this view, dreaming serves as a mechanism through which the brain processes and integrates newly acquired memories, gradually stripping away or reducing their emotional intensity[9]. In this light, large-scale stressors -such as the COVID-19 pandemic, which profoundly disrupted daily life on a global scale- could be expected to leave a significant imprint on dream content. Here we found that while narratives during the pandemic retained the immersive and hyper-associative characterization of ordinary dreams, they were more anchored into daily life dynamics and incorporated personal reflections typical of waking experiences. Notably, themes concerning healthcare, which were heavily represented in daily life during the pandemic, showed no significant changes. However, in a continuous line with what was happening in the daylight world, the actions of the individuals while they were dreaming were described as limited by physical or metaphorical constraints and the recalled emotional states carried a stronger intensity. This suggests that the pandemic modulated specific phenomenological dimensions, likely leading to changes in the hallucinatory depth and in the sense of immersion experienced by the dreamer. The analysis of longitudinal changes across the four-year following the pandemic's peak period revealed a progressive normalization of dream features, mirroring the epidemiological resolution of the global epidemic. Both dream and waking narratives demonstrated this restorative process, evolving toward more positive affective tones while showing decreasing pandemic-related thematic influence. This temporal pattern suggests that while significant stressors leave measurable imprints on dream phenomenology, these alterations appear to follow a recovering trajectory—diminishing as the psychological impact of the stressor wanes. Notably, this trend aligns with established findings showing parallel normalization of dream experiences and psychological symptoms following traumatic events[80,90].

### Limitations
Some limitations of this work should be acknowledged. First, the description of dreams as exhibiting narrative or cinematic structure arises from statistical regularities in the language of dream reports rather than from direct evidence about the underlying generative processes of dreaming[91]. Such structures may reflect recall and reporting biases, as participants tend to reconstruct their experiences using familiar narrative schemas. While the large-scale consistency of these linguistic patterns provides empirical evidence of internal organization, our methods cannot fully disentangle dream generation from narrative reconstruction during recall. Nevertheless, our direct comparison of waking and dream reports revealed state-specific differences, suggesting that these patterns are not solely attributable to reporting biases but may also reflect underlying differences in the experiences themselves. Second, although our longitudinal and cross-sample analyses revealed systematic associations between dream content, the COVID-19 pandemic, and the passage of time, these findings remain correlational. The observational nature of the data precludes any causal inference regarding the impact of external stressors or temporal factors on dream phenomenology.

### Conclusions
Together, these results bridge longstanding gaps between phenomenological dream research and cognitive neuroscience, offering testable hypotheses about the mechanisms linking dream content to memory consolidation, emotional regulation, and consciousness during sleep. The strong agreement between NLP-based semantic ratings and human evaluations—both from independent raters and from the dreamers themselves across repeated assessments (see Supplementary Figs. S1 and S2)—demonstrates that automated methods can reliably capture the nuances of phenomenological reports. At the same time, their scalability enables analyses that would be prohibitively time-consuming and inconsistent using traditional manual approaches. In the future, these automated approaches may enable longitudinal and cross-cultural comparisons of dreaming, reveal subtle links between dream features and psychological or neural states, and provide biomarkers for altered cognition in health and disease. By integrating advanced computational tools with established neuroscientific and psychological frameworks, dream research may thus move toward a more comprehensive and reproducible understanding of the mind's generative activity during sleep.

## Data availability
Preprocessed data to replicate these findings, to generate tables, and represent main and Supplementary Figs. are available at this link: https://zenodo.org/records/15230218.

## Code availability
MATLAB code to replicate these findings, to generate tables, and represent main and Supplementary Figs. are available at this link: https://zenodo.org/records/15230218.

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

## Acknowledgements

The authors thank Alessandro Lenci, Davide Bottari and Claudia Picard-Deland for their feedbacks on a preliminary version of the work and for providing constructive comments and suggestions which helped to improve the manuscript, Francesco Lomi, Luca Fuligni, Elena Capriglia, Monica Di Giuliano, Margherita Bozzoli, Federico Frau, Aurora Salina, Damiana Bergamo and Giulia Avvenuti for their help in data collection and preprocessing, and all volunteers for participating in this study. This work was supported by a grant from the BIAL Foundation (#091/2020) and by the European Union – Next Generation EU (Mission 4, Component 2, Investment 1.1; CUP D53D23009580006), project PRIN 2022 "The Language of Dreams: The Relationship Between Sleep Mentation, Neurophysiology, and Neurological Disorders" (2022BNE97C). In addition, this work was supported by the Resilienza Economica e Digitale project (CUP D67G23000060001), funded by the Italian Ministry of University and Research (MUR) under the Department of Excellence program (Dipartimenti di Eccellenza 2023–2027; Ministerial Decree no. 230/2022) (to G.H.), and by the TweakDreams ERC Starting Grant (#948891) (to G.Be.). The funders had no role in the study conceptualization and design, data collection, analysis, decision to publish, or preparation of the manuscript.

## Author contributions

Conceptualization: G.Be., M.B., G.H., V.E.; Investigation: V.E., G.Bo., B.P., S.S.; Methodology: G.H., G.Be., V.E.; Software: G.H., G.Be., V.E.; Formal analysis: V.E., G.H.; Visualization: V.E., G.Be., G.H.; Data curation: V.E., S.S., G.Bo.; Validation: V.E., G.Be., G.H.; Supervision: G.H., G.Be., P.P., L.D.; Funding acquisition: G.Be., M.B.; Project administration: G.Be.; Resources: G.Be., L.D., P.P; Writing—original draft: V.E., G.Be., G.H.; Writing—review and editing: All authors. All authors have read and agreed to the published version of the manuscript.

## Competing interests

The authors declare no competing interests.
