## [Transparent Peer Review file · Communications Psychology]

Individual Traits and Experiences Shape the Content of Dreams

Corresponding Author: Professor Giulio Bernardi

Version 0:

Decision Letter:

Dear Professor Bernardi,

Thank you for your patience during the peer-review process. Your manuscript titled "The semantics of dreams" has now been seen by 3 reviewers, whose comments are appended below. I have discussed the reports with my colleagues and I regret to inform you that we decided that in light of the referee reports, we cannot publish your manuscript in Communications Psychology.

You will see that the reviewers raise substantive concerns. Taking these points together with our editorial considerations, these reservations preclude publication of this study in Communications Psychology.

I am sorry that we cannot be more positive on this occasion and thank you for the opportunity to consider your work.

Best regards,

Troy Lui

Troy Lui, PhD
Associate Editor
Communications Psychology

REVIEWERS' EXPERTISE:

Reviewer #1: dream
Reviewer #2: dream
Reviewer #3: NLP

REVIEWER COMMENTS:

Reviewer #1 (Remarks to the Author):

Comments for Author

The manuscript "The Semantics of Dreams" presents a large-scale, computationally informed investigation of dream reports, linking their semantic organization to both individual traits and external events such as the COVID-19 pandemic.

Major claims and novelty

The paper makes the important claim that dream narratives are not only shaped by idiosyncratic experiences but also by stable, generalizable traits and shared external events. This is a valuable and timely contribution, since it bridges traditional phenomenological approaches with computational linguistics and natural language processing. The emphasis on large datasets and on cross-sample validation is an original strength, and the findings are likely to attract wide interest from both cognitive neuroscience and dream research communities.

Convincingness of the work

The narrative is generally clear, well-structured, and compelling. The use of NLP and large language models to assess dream semantics is innovative and well justified, and the discussion effectively situates the findings within broader theoretical frameworks (e.g., memory consolidation, emotion regulation, mind-wandering). At the same time, the study could benefit from acknowledging classical theoretical perspectives. For example, the discussion of dreams as “guardians of sleep” would be further enriched by explicitly referencing Freud’s formulation of this idea, which remains a cornerstone in the history of dream theory.

Methodology

I must note that I am not in a position to fully evaluate the methodological and statistical procedures employed in this study. For this reason, I will refrain from commenting on the technical robustness of the analyses. My review therefore focuses on the conceptual, theoretical, and interpretative aspects.

Potential influence on the field

The paper has the potential to become influential, especially in demonstrating how computational methods can advance the systematic study of dreams. The identification of semantic structures reminiscent of storytelling genres is particularly engaging and opens new avenues for interdisciplinary dialogue (e.g., with literary studies, cultural studies, and psychoanalysis).

Clarity and references

The writing is overall clear and accessible, though occasionally dense. It might help readers if the discussion more explicitly connected the empirical findings to existing debates in dream theory. Adding references to key frameworks, such as Revonsuo’s Threat Simulation Theory, would further strengthen the theoretical grounding.

Reproducibility and openness

The description of procedures, datasets, and computational approaches appears detailed and transparent, suggesting that the work is reproducible in principle. The inclusion of supplementary materials and code references is a valuable step toward open science practices.

Overall evaluation

In summary, this is an ambitious and stimulating paper that combines methodological innovation with theoretically significant insights. I recommend publication after minor revisions, mainly to temper certain claims and to strengthen the integration with foundational theories of dreaming, including Freud’s view of dreams as protectors of sleep.

Reviewer #2 (Remarks to the Author):

The manuscript aims to offer a comprehensive view of the themes within dreams as well as the general nature of dream content through exploratory analyses of dream content from a large sample across two main datasets. The title, the semantics of dreams, is a little unclear; I think it would be helpful both here and throughout the manuscript to articulate the aims and outcomes more explicitly. For instance, the analyses were exploratory (it seems) and observational in nature, though it would help to have a rationale for the investigations reported here, e.g. emotional features and states as documented in Fig. 1.

It would also help to understand how the analyses were nested by participant, as we know (and indeed your findings show) that individual differences impact on dream themes and states profoundly, so dreams from the same participant would almost certainly share more variance than dreams from different participants. How did the analytical models account for this?

The abstract needed to contain more conventional information about the approaches, participants and methods used. Critically, care should be taken to avoid implying causation of the COVID-19 pandemic upon dream characteristics from analyses of naturally-occurring observations.

I do think the findings need to be presented in a more conventional format and, as such, as typical empirical structure (with methods preceding findings, for instance) would help the reader to make sense of the complexities of the approach and exploratory analyses here.

Further discussion of the validity and potential utility of the AI models as part of the dream analysis process would be welcomed.

Although the findings are not entirely novel, they corroborate existing literature. The methods are thorough but the presentation is a little unclear; I would welcome a conventional presentation.

Reviewer #3 (Remarks to the Author):

Thanks for the chance to review this paper. Although the topic is certainly interesting, I do not think this paper is suitable for publication in this outlet. My main concern is the lack of theoretical motivation and integration, along with an unclear link to the methods chosen.

I realize the authors are following the journal format when preparing their article. However, I found the progression of the

paper difficult to follow in its current form. The following overarching comments would be useful to address. First, the Introduction did not clearly lay out which theoretical questions would be addressed; it felt a bit like a descriptive postdoc study. Second, I found it very difficult to read the results because there was a lack of statistics presented. Simply referring to the supplementary made the reading and main points hard to follow, especially given that the methodology was barely alluded to. In my experience, it is common to give important context in these early sections, even if the details are mostly in the Methods section.

You will see this theme throughout my review, but I am struggling to understand the broader contributions of the paper. If theoretical, I am not sure if the authors have done a convincing job of setting this up. The fact that people use different language to describe dreams and wakeful experiences is actually a pretty well known finding, even when comparing dreams to more “dream like” wake states such as mind wandering. The fact that dreams change as a result of our experiences is also not a very novel finding. Finally, the fact that there are “trait” level variables that predict dream reports is also not entirely novel and would need more much unpacking to integrate with theory more deeply. If methodological, I think the authors would need to do much more to compare to other methods and why these are “better.”

The Introduction was a bit unsatisfying in terms of deeply connecting to theory and prior studies, especially with respect to motivating core research questions that are accompanied by clear hypotheses.

The NLP methods seem like they are solid and accurately applied. My question here is not about the specific methods used but more so why the authors chose these approaches in the first place. For example, why did the authors think there would be different domains and why is this theoretically important to compare across wake and dreams as well as over time? Specifically, why is this approach the best way to answer the broader theoretical question?

In general, I found the methods difficult to follow because it was not always clear why certain choices were made and how they mapped to specific research questions.

I am especially confused when the authors get into things like polynomial models because this was not motivated at all by a theoretical reason.

The models reported have a complex structure with several control variables. This is a good thing, but would also like to see if the models are similar without controlling for all of this variance.

The dimensions chosen were motivated by previous literature for the LLM coding, but I don't think this was unpacked very well. Why are these, of all possible dimensions, the ones that are most representative of dreams and wakefulness, especially as it would relate to the core hypotheses?

How were the trait level measures chosen? This is important and seems like it is missing a theoretical motivation or deep interpretation that helps theory.

There is an assumption that dreams themselves follow the narrative structure of a movie, etc. This is one *possible* interpretation of the data, but it is not exactly founded in the methods used here. First, it could be based on the methods, such that the recall of dreams is constrained by the way recall them, which might itself fit with schemas of narratives we experience in our lives like movies etc. Second, this is a claim that is somewhat unfalsifiable using the methods here. What is the null of the question being tested along these lines?

The fact that “trait level” variables predicted aspects of the dreams is once again a bit of a confusing interpretation. The idea that this is not at least partially accounted for in the methods is not deeply considered. I realize this is a limitation of much of the work done with recalls, but my concern is that the paper is making claims based on the fact that some people were more interested in their dreams, which seems like an obvious confound rather than a theoretical contribution.

The downsampling approach is a bit confusing. It seems like the 13% of random sampling could have affected results. It might be nice to check for results when that 13% is not considered.

How did the authors confirm that their word tagging was accurate if it was done automatically?

Note on appeals: In exceptional circumstances, it is in authors' interest to appeal an editorial decision. More information on appeals is available here: <https://www.nature.com/commmspsychol/submit/editorial-process#appeals>

Version 1:

Decision Letter:

Dear Professor Bernardi,

Thank you for your correspondence asking us to reconsider our decision on your Article, "The semantics of dreams". After careful consideration we have decided that we would be willing to consider a full appeal in the form of a revised version of your manuscript.

As you note in your letter, there are substantive methodological concerns that you will need to comprehensively address. In addition, the decision to reject the work following the previous round of reviewer reports also reflected that both Reviewer #2 and Reviewer #3 voiced critical concerns regarding the advance the work presents and the cognitive framework underlying the study. The revised manuscript must address these issues, which involves clarifying exactly how the work confirms, extends, or contradicts previous findings. That the work is exploratory and phenomenological in nature rather than a strict test of theory or mechanism should be made explicit in the revised manuscript.

Along with your revised manuscript, you should also submit a separate point-by-point response to all of the concerns raised by the referees, in each case describing what changes have been made to the manuscript.

Please note that we will only take the appeal forward and contact the reviewers again if we are persuaded that a substantial attempt has been made to address all editorial concerns and referees' comments. In this case, your revised manuscript and the point-by-point reply will be sent back to the referees so that they can judge whether their concerns have been addressed satisfactorily or otherwise.

I should stress, however, that we would be reluctant to trouble our referees again unless we thought that their comments had been addressed in full.

When revising your paper please:

- ensure that it complies with the editorial policies outlined below
- ensure it meets our format requirements as set out in our [Guide to Authors](http://www.nature.com/commspsychol/submit/guide-to-authors) and highlighted below.
- ensure that the statistics reporting and interpretation is in line with journal guidelines <https://www.nature.com/commspsychol/submit/submission-guidelines#statistical-guidelines>

Please mark all correspondence via email with your Communications Psychology reference number in the subject line.

If the revision process takes significantly longer than five months, we will be happy to reconsider your paper at a later date, provided it still presents a significant contribution to the literature at that stage.

Please use the following link to submit your revised manuscript, point-by-point response to the Reviewers' comments with a list of your changes to the manuscript text (which should be in a separate document to any cover letter) and any completed checklist:

Link Redacted

Best regards,

Troy Lui

Troy Lui, PhD
Associate Editor
Communications Psychology

EDITORIAL POLICIES AND FORMATTING

Furthermore, please align your manuscript with our format requirements, which are summarized on the following checklist: <https://www.nature.com/documents/commspsychol-style-formatting-checklist-article-rr.pdf> Communications Psychology formatting checklist

and also in our style and formatting guide Communications Psychology formatting guide .

Version 2:

Decision Letter:

Dear Professor Bernardi,

Your manuscript titled "The semantics of dreams" has now been seen by our reviewers, whose comments appear below. In light of their advice I am delighted to say that we are happy, in principle, to publish a suitably revised version in Communications Psychology.

We therefore invite you to revise your paper one last time to address the remaining concerns of our reviewers and a list of editorial requests. At the same time we ask that you edit your manuscript to comply with our format requirements and to maximise the accessibility and therefore the impact of your work.

EDITORIAL REQUESTS:

SUBMISSION INFORMATION:

In order to accept your paper, we require the files listed here <https://www.nature.com/documents/commsj-file-checklist.pdf> .

OPEN ACCESS:

* DATA AVAILABILITY:

All Communications Psychology manuscripts must include a section titled "Data Availability" at the end of the Methods section. More information on this policy, is available in the Editorial Requests Table and at ><http://www.nature.com/authors/policies/data/data-availability-statements-data-citations.pdf>.

Link Redacted

Best regards,

Troby Lui

Troby Lui, PhD
Associate Editor
Communications Psychology

REVIEWERS' COMMENTS:

Reviewer #1 (Remarks to the Author):

I thank the authors for their careful and substantial revision of the manuscript. The paper has clearly improved in clarity, structure, and theoretical framing, and the revisions demonstrate a serious engagement with the reviewers' comments.

In particular, my previous concerns regarding the integration of classical and contemporary theories of dreaming, as well as the positioning of the work within existing debates on continuity, emotion regulation, and individual traits, have been adequately addressed. The revised Introduction and Discussion now provide a clearer rationale for the analytical approach and a more balanced interpretation of the findings.

Overall, I believe the manuscript now makes a solid and coherent contribution to the literature and is suitable for publication. I have no further major comments.

Reviewer #3 (Remarks to the Author):

I appreciate the opportunity to re-review this manuscript. On the first review, I have to admit that I was quite negative and recommended rejection. However, the authors have done a really excellent job on their revision — both in terms of their response letter and the updates to the manuscript. I can now see a very clear path to publication and think this work has the potential to contribute to the field.

My remaining comments/concerns are below, but they are relatively minor in nature.

First, I think the Introduction is substantially better, so I appreciate the responsiveness. At the same time, I think the authors have kept the Intro too brief at the expense of being able to connect to prior literature at a deeper level and be able to make stronger predictions. I understand the aims are laid out nicely, but I think it still says what was examined and not why it is important to understand in terms of producing theoretically-novel results. I would still recommend moving away from the incredibly brief report style of writing and going a bit deep. This is especially true for 1) being able to make predictions and 2) specifically motivating the key variables/constructs that were chosen to include. Right now they are mentioned and feel more or less like a list of things that the authors are assuming should be included. (Note, the authors do now mention these measures very briefly, but I do think some of this — especially the theoretical ties — should be connected to the theories in the Intro. Shorter version of this comment: in general, some of the mention of the measures is still feeling a bit shallow and I'd like to see that addressed.

Second, I'm still not totally convinced by the LLMs scoring methods, so I'd like to see more in the manuscript — not the Supplementary Material. For example, the paper says that there was correlations on average of .6. What was the distribution? .6 is actually not very high in general, and if some were quite a bit lower, then should we trust those dimensions?

Third, in terms of the discussion. I still there there could be more work to go deeper here too. Part of the issue is that there are so many variables and so many results, which makes the overall discussion a bit difficult to make cohesive -- but do think this is possible.

I commend the authors for adding more linkages to prior work compared to the last version, but it still feels like some important work is missed and some constructs are simply not incorporated. Sometimes citations are missing for discussions that have already taken place in the literature (e.g., dreaming being a form of mind wandering; see Christoff et al., 2016; Mallet et al., 2025), as well as other examples that I will not exhaustively detail. Perhaps most importantly, the discussion still doesn't tie very well with the Intro (see comments above). My suggestion is to deepen both in ways that will also make them more cohesive in terms of the goals and contribution to theory.

I really appreciate the limitations paragraph. This is really useful to have added.

December 27th, 2025

We sincerely thank the Editor and the Reviewers for the opportunity to resubmit our manuscript, as well as for their thoughtful and constructive feedback throughout this process. We have carefully considered all comments and substantially revised the manuscript to address the concerns raised in the initial evaluation, especially regarding the clarity of the manuscript, its positioning within the existing literature, and the articulation of the novelty and rationale of our analyses.

In preparing this revised version, we undertook an extensive restructuring and rewriting of the manuscript, with major revisions across more than half of the text. While the core analyses, results, and conclusions remain unchanged, we have significantly strengthened the theoretical framing, clarified the contribution of our work relative to prior studies, and provided a more explicit rationale for our analytical approach. We also improved the presentation and interpretation of the findings in light of current theories and empirical literature on the origin and significance of dreaming.

We hope that these substantial revisions address the Reviewers' concerns and improve the clarity, coherence, and overall contribution of the manuscript. Below, we provide a detailed, point-by-point response to each comment and a description of the changes implemented.

Reviewer #1

The manuscript “The Semantics of Dreams” presents a large-scale, computationally informed investigation of dream reports, linking their semantic organization to both individual traits and external events such as the COVID-19 pandemic. [...] The paper makes the important claim that dream narratives are not only shaped by idiosyncratic experiences but also by stable, generalizable traits and shared external events. This is a valuable and timely contribution, since it bridges traditional phenomenological approaches with computational linguistics and natural language processing. The emphasis on large datasets and on cross-sample validation is an original strength, and the findings are likely to attract wide interest from both cognitive neuroscience and dream research communities.

We thank the Reviewer for their positive assessment of our work and their constructive feedback.

Convincingness of the work. The narrative is generally clear, well-structured, and compelling. The use of NLP and large language models to assess dream semantics is innovative and well justified, and the discussion effectively situates the findings within broader theoretical frameworks (e.g., memory consolidation, emotion regulation, mind-wandering). At the same time, the study could benefit from acknowledging classical theoretical perspectives. For example, the discussion of dreams as “guardians of sleep” would be further enriched by explicitly referencing Freud’s formulation of this idea, which remains a cornerstone in the history of dream theory.

We thank the Reviewer for this valuable suggestion. We fully agree that integrating classical theoretical perspectives enriches the conceptual framing of the study. Accordingly, we have expanded the Introduction and Discussion sections to better place our present work in the context of the existing literature and theories about dreaming.

Clarity and references. The writing is overall clear and accessible, though occasionally dense. It might help readers if the discussion more explicitly connected the empirical findings to existing debates in dream theory. Adding references to key frameworks, such as Revonsuo’s Threat Simulation Theory, would further strengthen the theoretical grounding.

We extensively reviewed the manuscript to improve overall clarity and better explain the rationale and context of each analysis. As mentioned above, we also expanded the Introduction and Discussion section to better highlight aspects of novelty and the relationship between our results and existing theories concerning the nature and function of dreams.

Overall evaluation. In summary, this is an ambitious and stimulating paper that combines methodological innovation with theoretically significant insights. I recommend publication after minor revisions, mainly to temper certain claims and to strengthen the integration with foundational theories of dreaming, including Freud’s view of dreams as protectors of sleep.

We thank the Reviewer for this positive assessment and sincerely hope the applied changes contribute to improve the manuscript’s quality.

Reviewer #2

The manuscript aims to offer a comprehensive view of the themes within dreams as well as the general nature of dream content through exploratory analyses of dream content from a large sample across two main datasets. The title, the semantics of dreams, is a little unclear; I think it would be helpful both here and throughout the manuscript to articulate the aims and outcomes more explicitly. For instance, the analyses were exploratory (it seems) and observational in nature, though it would help to have a rationale for the investigations reported here, e.g. emotional features and states as documented in Fig. 1.

We thank the Reviewer for this helpful comment. We agree that the aims and rationale of the study should be stated more explicitly, and we have revised both the title and Introduction accordingly. The revised title, “*The Semantics of Dreams: how individual Traits and Experiences shape Dream Content*”, now more clearly reflects the study’s scope and objectives.

In the Introduction, we now explicitly describe the rationale and hypotheses underlying each analytical step, clarifying that the work adopts an exploratory yet hypothesis-guided approach. Specifically, we outline three main aims: (i) to characterize the similarities and differences between dream and waking reports (continuity vs. distinctiveness of cognition across vigilance states); (ii) to examine how stable individual traits shape dream content; and (iii) to assess the influence of external events and temporal factors on the semantics of dreams. These additions are presented in the revised Introduction (pp. 3-4).

We have also added brief clarifying statements in the Results and Methods sections explaining the rationale for each analysis to ensure that aims and outcomes are clearly aligned throughout the manuscript.

It would also help to understand how the analyses were nested by participant, as we know (and indeed your findings show) that individual differences impact on dream themes and states profoundly, so dreams from the same participant would almost certainly share more variance than dreams from difference participants. How did the analytical models account for this?

We apologize for the lack of clarity concerning this point. Indeed, dreams from the same participant are not statistically independent, and this within-subject dependency was explicitly accounted for in all our analyses. Specifically, we employed Generalized Linear Mixed-Effects (GLME) models that included *participant* as a random effect. This approach allowed us to model intra-individual variance across repeated reports and to isolate effects that are consistent across participants from those driven by individual idiosyncrasies. For each analysis (e.g., comparisons between vigilance states, trait effects, and temporal trends), the corresponding model specification using Wilkinson’s notation included the term ‘(1 | Participant)’, ensuring that all statistical inferences reflected variability both within and between participants. We have clarified this point explicitly in the revised Methods section (see Materials and Methods, p. 13).

The abstract needed to contain more conventional information about the approaches, participants and methods used. Critically, care should be taken to avoid implying causation of the COVID-19 pandemic upon dream characteristics from analyses of naturally-occurring observations.

We thank the Reviewer for these constructive suggestions. We have revised the Abstract to include more conventional methodological information, specifying the sample size, datasets, duration of data collection, and the main analytical approaches. The revised abstract now clearly outlines the three main aims of the study, the methods applied, and the core findings (see revised Abstract).

We have also carefully revised the wording throughout the Abstract, Results, and Discussion to ensure that no causal interpretation is implied regarding the relationship between the COVID-19 pandemic and changes in dream characteristics. The text now explicitly states that our analyses are correlational and intended to identify associations rather than causal effects (see Abstract, lines 48-51; Results, pp. 19-27).

I do think the findings need to be presented in a more conventional format and, as such, as typical empirical structure (with methods preceding findings, for instance) would help the reader to make sense of the complexities of the approach and exploratory analyses here.

We thank the Reviewer for this valuable suggestion. In line with this recommendation, we have restructured the manuscript to follow a more conventional empirical format, with the Methods section now preceding the Results. This reorganization improves the logical flow and helps readers better follow the progression from study design and analytical rationale to the presentation of findings.

In addition, we refined the internal structure of both sections to further enhance clarity. Each Results subsection now begins with a concise statement of the analytical objective and references the corresponding methodological details to facilitate cross-reading. We believe this revised organization makes the manuscript clearer and more accessible, particularly given the methodological complexity of the analyses.

Further discussion of the validity and potential utility of the AI models as part of the dream analysis process would be welcomed.

We have expanded the Discussion to include a dedicated reflection on the validity and future potential of AI-based approaches in dream research. Specifically, we now highlight that the strong agreement between AI-generated semantic ratings and human evaluations (both from independent raters and from participants evaluating their own dreams) supports the reliability and interpretive accuracy of the NLP models used in this study. We further discuss the broader utility of AI tools for large-scale, reproducible analyses of subjective experiences, emphasizing how these methods can bridge qualitative and quantitative approaches and enable future longitudinal and cross-cultural investigations of dreaming. These additions are presented in the revised Discussion (pp. 31-32, lines 1903–1103).

Although the findings are not entirely novel, they corroborate existing literature. The methods are thorough but the presentation is a little unclear; I would welcome a conventional presentation.

In light of this and comments from the other Reviewers, we extensively revised the manuscript and hope that this new version may be clearer and more accessible. Regarding the comment on novelty, we would like to emphasize that, while our results corroborate prior findings on the general influence of waking experience on dreaming, they also provide a quantitative and reproducible mapping of how stable individual

traits and shared external events shape the semantic organization of dreams, an aspect that has not previously been examined at this scale or level of detail. This combination of large-scale natural language modeling and multimodal individual characterization represents a methodological and conceptual advance that extends the existing literature.

Reviewer #3

Thanks for the chance to review this paper. Although the topic is certainly interesting, I do not think this paper is suitable for publication in this outlet. My main concern is the lack of theoretical motivation and integration, along with an unclear link to the methods chosen.

We thank the Reviewer for their feedback and appreciate the opportunity to clarify how the revised version of the manuscript now more explicitly links our theoretical rationale to the methodological approach. We have substantially expanded the Introduction and Discussion to strengthen the theoretical motivation and integration of the study. These revisions make explicit how our analyses are grounded in long-standing debates on continuity and transformation between waking and dreaming cognition. We also clarify the link between theory and methods: each analytical step (e.g., the use of natural language processing, the comparison of dream and waking reports, and the modeling of individual traits) directly addresses theoretical questions about the mechanisms shaping dream phenomenology, such as whether dreams preserve or reorganize waking experience, and how individual and external factors modulate this process. These revisions ensure that the theoretical rationale is now explicit, cohesive, and clearly aligned with the analytical design, making the study's motivation and methodological choices transparent to the reader. Please further see our point-by-point response to each specific comment.

I realize the authors are following the journal format when preparing their article. However, I found the progression of the paper difficult to follow in its current form. The following overarching comments would be useful to address. First, the Introduction did not clearly lay out which theoretical questions would be addressed; it felt a bit like a descriptive postdoc study. Second, I found it very difficult to read the results because there was a lack of statistics presented. Simply referring to the supplementary made the reading and main points hard to follow, especially given that the methodology was barely alluded to. In my experience, it is common to give important context in these early sections, even if the details are mostly in the Methods section.

We thank the Reviewer for these constructive comments and have implemented several major revisions to improve the manuscript's clarity, structure, and accessibility.

1) Theoretical motivation. The Introduction has been rewritten to clearly articulate the study's rationale and theoretical grounding. It now outlines three main aims: (i) comparing dreaming and waking experiences to test continuity and discontinuity, (ii) examining the influence of stable individual traits on dream content, and (iii) assessing how external events and temporal factors modulate dream semantics, and integrates these aims within both classical and contemporary frameworks (pp. 3-4).

2) Statistical reporting and clarity. We would like to clarify that including all statistical details in the main text would have been impractical given the large number and complexity of the tested models. To ensure

both transparency and readability, we reported in the main text and figures only statistically significant results after FDR correction ($q < 0.05$), together with the corresponding effect sizes, which are essential for evaluating the strength and practical relevance of the observed effects beyond their probabilistic significance. Additional statistical information (including non-significant effects) is provided in the figures and figure legends, and all comprehensive outputs remain available in the Supplementary Information and in the ‘derivatives/’ folder alongside the shared data and code. In the revised manuscript, we have clarified this reporting rationale and, to further enhance transparency and accessibility, have moved the most interpretable and relevant tables and figures from the Supplementary Information to the main text. In particular, we moved to the main text the tables reporting the performance measures of the GLME models for the prediction of semantic dimensions and lexical domains across vigilance states (pp. 22 - 24, Tables 1 - 3), as well as the table reporting the absolute frequencies of lexical domains in the main and lockdown datasets (p. 23, Table 2).

3) Improved flow and contextualization. The manuscript has been restructured so that the Methods section now precedes the Results, consistent with the Reviewer’s suggestion and the journal’s final publication format. In addition, we have inserted short methodological reminders at the start of each Results subsection, summarizing the analytical approach to help readers follow the rationale behind each analysis without referring back to the Methods.

We believe these revisions substantially improve the manuscript’s theoretical integration, readability, and statistical transparency, making the results clearer and more interpretable to a broad interdisciplinary audience.

You will see this theme throughout my review, but I am struggling to understand the broader contributions of the paper. If theoretical, I am not sure if the authors have done a convincing job of setting this up. The fact that people use different language to describe dreams and wakeful experiences is actually a pretty well known finding, even when comparing dreams to more “dream like” wake states such as mind wandering. The fact that dreams change as a result of our experiences is also not a very novel finding. Finally, the fact that there are “trait” level variable that predict dream reports is also not entirely novel and would need more much unpacking to integrate with theory more deeply. If methodological, I think the authors would need to do much more to compare to other methods and why these are “better.”

We appreciate the Reviewer’s comments and the opportunity to clarify the novelty and broader contribution of our study.

While previous research has shown that certain individual traits influence dream recall frequency, evidence on how such traits shape dream content itself, and the organization of its semantic features, remains extremely limited. Our work represents the first large-scale, quantitative, and systematic mapping of how multiple stable individual traits and shared external events jointly influence dream content, using validated NLP methods and cross-dataset replication to ensure robustness.

Importantly, our study moves beyond confirming that dreams are shaped by experience: it provides a first empirical and statistically grounded exploration of how these influences may manifest, evolve over time,

and differ between individuals. This includes the first longitudinal characterization of semantic changes in dream content across four years following the pandemic.

Finally, our focus was not on surface-level linguistic variation but on semantic and conceptual organization, the thematic and structural properties of dream narratives, addressing a question that has not been systematically quantified in prior literature.

A further, though not primary, contribution of the work lies in demonstrating the applicability and reliability of modern NLP tools in dream research, a field that has traditionally relied on manual scoring or simple word-counting approaches. Through thorough validation against both independent human raters and self-ratings by the dreamers themselves, we show that these computational methods can capture the semantic structure of subjective reports with human-level accuracy.

These aspects, which are now made more explicit in the revised manuscript (Introduction, pp. 3-4; Discussion, pp. 31-32), together establish the study as both conceptually novel and methodologically innovative within the field of dream research.

The Introduction was a bit unsatisfying in terms of deeply connecting to theory and prior studies, especially with respect to motivating core research questions that are accompanied by clear hypotheses.

We thank the Reviewer for this helpful feedback. In the revised manuscript, we have substantially expanded and reorganized the Introduction to strengthen its theoretical integration and clarify the specific research questions addressed. We also now explicitly articulate the three guiding hypotheses of the study: i) That dream and waking experiences share common semantic structures but differ systematically in key dimensions, such as affective and perceptual characterization; ii) That individual traits selectively influence dream content beyond their effects on language or narrative style; and iii) That large-scale external events and temporal factors are associated with changes in dream semantics over time. These revisions clarify both the theoretical grounding and empirical aims of the work, ensuring that the research questions are explicitly motivated by prior studies and positioned within ongoing theoretical debates (see revised Introduction, pp. 3-4).

The NLP methods seem like they are solid and accurately applied. My question here is not about the specific methods used but more so why the authors chose these approaches in the first place. For example, why did the authors think there would be different domains and why is this theoretically important to compare across wake and dreams as well as over time? Specifically, why is this approach the best way to answer the broader theoretical question?

We adopted two complementary NLP frameworks, hypothesis-driven semantic dimensions and data-driven lexical domains, because they capture distinct yet complementary aspects of dream and waking mentation. The semantic-dimension approach was explicitly hypothesis-driven, grounded in previous research on dreaming based on manual content ratings, and allowed us to quantify key features traditionally emphasized in the literature (e.g., perceptual vividness, emotional tone, social content, and agentivity) in a reproducible and scalable way. The lexical-domain approach, in contrast, was data-driven and designed to identify

recurring themes and semantic clusters that are expected to emerge in any form of subjective experience, without relying on prior theoretical constraints.

For example, the dream report “*I was in a restaurant and then, let’s say, after having eaten I received, precisely, the bill, but the bill was an amount that was very high, disproportionate, and therefore, well, uh, I was worried and anxious for having spent too much.*” shows, on semantic dimensions, a lower *Valence* score (2 on a 1–9 scale, with 4 indicating neutrality) and higher *Arousal* and *Body*-related reference scores compared with the average across all dream reports. By contrast, analysis of lexical domains indicates that the report contains a higher proportion of references to the *Food*, *Reactions*, and *Transaction* classes than the average probability distribution observed across all reports.

These two methods were therefore chosen for their complementarity. Comparing these measures across vigilance states and time directly addresses core theoretical questions about whether dreaming represents a continuous extension of waking cognition or a qualitatively distinct mode of experience, and how such organization evolves under major societal or emotional perturbations. This integrated framework represents, to our knowledge, the most comprehensive approach to date for quantifying the continuity and transformation of semantic structures across vigilance states (dream vs. wakefulness) and over time, thereby addressing central theoretical questions about the organization and function of dreaming.

We have substantially expanded the methods section to explicitly articulate the rationale behind each analytic choice, including the theoretical justification for the selected semantic dimensions and the complementary role of the data-driven approach (Introduction, pp. 3-4, Materials and Methods, pp. 9-18).

In general, I found the methods difficult to follow because it was not always clear why certain choices were made and how they mapped to specific research questions.

We thank the Reviewer for this helpful comment. We have thoroughly revised the Methods and Results sections to make the rationale behind each analytical step clearer and to explicitly link methodological choices to the corresponding research questions and hypotheses. These changes ensure that the methods-to-question mapping is now explicit and that the manuscript’s structure guides the reader through the rationale, analytical design, and corresponding results.

I am especially confused when the authors get into thing like polynomial models because this was not motivated at all by a theoretical reason.

We thank the Reviewer for this comment and would like to clarify the possible sources of confusion.

If the Reviewer was referring to the mixed-effects models including multiple predictors and interactions, these were implemented intentionally to account for participant-related variance and shared variance among correlated predictors, thereby isolating the specific contribution of each variable. This approach reduces the risk of spurious findings and improves interpretability, as several traits (e.g., anxiety, sleep quality, and mind-wandering) are interrelated and may each appear significant when tested in isolation. We have now made this rationale more explicit in the Methods (pp. 13).

Alternatively, if the Reviewer was referring to the polynomial fitting mentioned in the figure captions and Methods, we clarify that polynomial functions were not used in the statistical analyses, but only for trend visualization in Fig. 3B–C. Specifically, these polynomial fits (up to the third degree) were applied post hoc to the adjusted data to illustrate the temporal evolution of the effects identified in the models. We have now revised the relevant section of the Methods to make this explicit and prevent misinterpretation (see below):

“The effect of the passage of time on the adjusted variables was visualized in Fig. 3B–C. For illustrative purposes only, we fitted the optimal polynomial function (up to the third degree) to the adjusted data for wakefulness and dream reports independently when a significant interaction with time was detected (Fig. 3B). For main effects of time, the entire dataset was aggregated prior to trend estimation (Fig. 3C). These polynomial fits were used solely for visualization and did not contribute to the statistical modeling or inference reported in the results.”

The models reported have a complex structure with several control variables. This is a good thing, but would also like to see if the models are similar without controlling for all of this variance.

As noted in our response above, our main models included multiple covariates (sex, age, education, and verbosity) to control for potential confounding effects and to isolate the specific contribution of each predictor. To address the Reviewer’s request, we have now verified that the pattern and direction of the main effects remain consistent when the models are re-run without these control variables (see Table R1-4, below).

The key results, including the direction and statistical significance of the main findings, were qualitatively unchanged, confirming that the observed effects are robust to model simplification. Notably, of the 62 tests performed, all critical effects remained stable when these covariates were removed, except for one (1.6% of all tests), marked with * in the tables below. We have added a note in the Methods (pp. 13-18) and Results (pp. 25) clarifying that control variables were included to account for potential demographic and linguistic confounds.

Table R1

Dimension	report-type coefficient	q-value coefficient	Results based on Reviewer's request	
			report-type coefficient	non-par p
AGENTIVITY	-1.58, CI: -1.70 -1.46	< 0.00001	-1.578	p<0.0001
AROUSAL	0.21, CI: 0.10 0.31	0.01216	0.206	p=0.0002
AUDITORY	0.42, CI: 0.30 0.55	< 0.00001	0.421	p<0.0001
BIZARRENES	2.85, CI: 2.73 2.98	< 0.00001	2.853	p<0.0001
BODY	-0.73, CI: -0.85 -0.61	< 0.00001	-0.733	p<0.0001
LIMITATIONS	0.82, CI: 0.69 0.95	< 0.00001	0.822	p<0.0001
THOUGHT	-1.77, CI: -1.88 -1.66	< 0.00001	-1.770	p<0.0001
MOVEMENTS	0.83, CI: 0.70 0.96	< 0.00001	0.833	p<0.0001
SETTINGS	1.29, CI: 1.18 1.40	< 0.00001	1.293	p<0.0001
SOCIAL	1.40, CI: 1.22 1.58	< 0.00001	1.399	p<0.0001
SPACE	1.40, CI: 1.30 1.51	< 0.00001	1.404	p<0.0001
TACTILE	0.49, CI: 0.40 0.58	< 0.00001	0.492	p<0.0001
TIME	-1.20, CI: -1.33 -1.08	< 0.00001	-1.204	p<0.0001
VALENCE	-0.49, CI: -0.59 -0.38	< 0.00001	-0.488	p<0.0001
VISUAL	1.52, CI: 1.38 1.66	< 0.00001	1.520	p<0.0001

Table R1. Comparison of the GLME model performance for the prediction of semantic dimensions across vigilance states, with and without the inclusion of covariates of no interest. In the main analysis, vigilance state (report-type coefficient) was entered as the regressor of interest, while age, sex, education level, and BADA score were included as covariates of no interest. The last two columns report the estimate of the report-type coefficient and its non parametric p-value (number of permutations = 5000) for each semantic dimension surviving FDR correction when covariates are excluded. FDR: False Discovery Rate, $q < 0.05$.

Table R2

Domain	report-type coefficient	q-value coefficient	Results based on Reviewer's request	
			report-type coefficient	non-par p
DOMAIN2	-0.53, CI: -0.75 -0.32	0.00008	-0.536	p<0.0001
DOMAIN4	-0.53, CI: -0.69 -0.37	< 0.00001	-0.528	p<0.0001
Food	-0.37, CI: -0.56 -0.18	0.00165	-0.366	p=0.0002
Objects	0.80, CI: 0.57 1.02	< 0.00001	0.768	p<0.0001
Reactions	-0.21, CI: -0.39 -0.02	0.04289	-0.207	p=0.0292
DOMAIN12	0.50, CI: 0.24 0.75	0.00008	0.489	p=0.0002
Matter	0.56, CI: 0.31 0.82	< 0.00001	0.559	p<0.0001
Thriller	0.99, CI: 0.79 1.19	< 0.00001	0.978	p<0.0001
Healthcare	-1.17, CI: -1.38 -0.97	< 0.00001	-1.160	p<0.0001
Locations	0.84, CI: 0.67 1.02	< 0.00001	0.844	p<0.0001
Geometry	1.00, CI: 0.81 1.19	< 0.00001	0.997	p<0.0001
Nature	0.48, CI: 0.32 0.64	< 0.00001	0.480	p<0.0001
Fantasy	-0.32, CI: -0.55 -0.08	0.01335	-0.315	p<0.0001
Measurements	-0.56, CI: -0.82 -0.30	< 0.00001	-0.557	p<0.0001
Architecture	1.08, CI: 0.86 1.29	< 0.00001	1.085	p<0.0001
Animals	0.93, CI: 0.72 1.14	< 0.00001	0.928	p<0.0001
Concerns	-0.62, CI: -0.79 -0.44	< 0.00001	-0.611	p<0.0001
Appearance	1.13, CI: 0.92 1.34	< 0.00001	1.132	p<0.0001
Jobs	0.97, CI: 0.80 1.14	< 0.00001	0.969	p<0.0001
Drama	-0.25, CI: -0.40 -0.11	0.01335	-0.253	p<0.0001
Society	0.48, CI: 0.30 0.66	< 0.00001	0.482	p<0.0001
Humanities	-1.20, CI: -1.39 -1.02	< 0.00001	-1.203	p<0.0001
Transportation	0.51, CI: 0.32 0.70	< 0.00001	0.506	p<0.0001
Timing	-1.54, CI: -1.69 -1.39	< 0.00001	-1.538	p<0.0001
Technology	-0.86, CI: -1.07 -0.65	< 0.00001	-0.852	p<0.0001
Education	-0.47, CI: -0.63 -0.32	< 0.00001	-0.473	p<0.0001
Communication	-0.46, CI: -0.62 -0.30	< 0.00001	-0.461	p<0.0001
Slang	-0.46, CI: -0.73 -0.20	0.00144	-0.461	p=0.0006

Table R2. Comparison of the GLME model performance for the prediction of lexical domains across vigilance states, with and without the inclusion of covariates of no interest. In the main analysis, vigilance state (report-type coefficient) was entered as the regressor of interest, while age, sex, education level, and BADA score were included as covariates of no interest. The last two columns report the estimate of the report-type coefficient and its non parametric p-value (number of permutations = 5000) for each domain surviving FDR correction when covariates are excluded. FDR: False Discovery Rate, $q < 0.05$.

Table R3

Dimension	coefficient name	coefficient beta	q-value coefficient	Results based on Reviewer's request	
				coefficient beta	par p
AROUSAL					
	report_type_1:ATD	0.05, CI: 0.02 0.08	0.04492	0.037	p=0.0102
BIZARRENESS					
	report_type_1:PSQI	-0.09, CI: -0.15 -0.03	0.02496	-0.080	p=0.0076
	report_type_1:ATD	0.07, CI: 0.03 0.10	0.00296	0.055	p=0.0011
	report_type_1:MW	0.27, CI: 0.16 0.38	0.00005	0.271	p<0.0001
	report_type_1:SCWT	-0.03, CI: -0.05 -0.01	0.04665	-0.018	p=0.0414
SETTINGS					
	report_type_1:MW	0.23, CI: 0.14 0.33	0.00008	0.235	p<0.0001
SPACE					
	report_type_1:ATD	0.07, CI: 0.04 0.10	0.00017	0.065	p<0.0001
VISUAL					
	report_type_1:ATD	0.07, CI: 0.03 0.10	0.02136	0.054	p=0.0038

Table R3. Comparison of the GLME model performance for the prediction of semantic dimensions based on individual psychological variables. In the main analysis, the GLME model included psychological variables and their interaction with vigilance states (*report_type*) as regressors of interest and age, sex, education level, and the BADA score as covariates. The last two columns report the estimate of the coefficients and the parametric *p*-value for each dimension surviving FDR correction when covariates are excluded. FDR: False Discovery Rate, $q < 0.05$.

Table R4

	Domain	coefficient name	coefficient beta	q-value coefficient	Results based on Reviewer's request	
					coefficient beta	par p
	Objects					
		ROCFr	-0.07, CI: -0.11 -0.03	0.00564	-0.061	p=0.0011
		report_type_1:ROCFr	0.09, CI: 0.04 0.14	0.00564	0.082	p=0.0005
	Matter					
		PSQI	-0.18, CI: -0.30 -0.06	0.03306	-0.206	p=0.0005
		report_type_1:PSQI	0.22, CI: 0.08 0.36	0.03306	0.236	p=0.0007
	Geometry					
		report_type_1:ATD	0.09, CI: 0.04 0.15	0.01627	0.091	p=0.0004
	Nature					
		report_type_1:ATD	0.07, CI: 0.03 0.12	0.04748	0.063	p=0.0032
	Appearance					
		report_type_1	-5.82, CI: -9.87 -1.76	0.04854	-2.305	p=0.1079*
		PSQI	-0.15, CI: -0.26 -0.05	0.04854	-0.166	p=0.0025
		report_type_1:PSQI	0.17, CI: 0.06 0.29	0.04854	0.179	p=0.0019
	Communication					
		report_type_1:MEQ	0.03, CI: 0.01 0.04	0.04769	0.029	p=0.0009
	Slang					
		report_type_1:MW	-0.39, CI: -0.63 -0.15	0.01454	-0.373	p=0.0023

Table R4. Comparison of the GLME model performance for the prediction of lexical domains based on individual psychological variables. In the main analysis, the GLME model included psychological variables and their interaction with vigilance states (report_type) as regressors of interest and age, sex, education level, and the BADA score as covariates. The last two columns report the estimate of the coefficients and the parametric p-value for each domain surviving FDR correction when covariates are excluded. FDR: False Discovery Rate, $q < 0.05$. * denotes parametric p values that after the removal of covariates resulted > 0.05 .

The dimensions chosen were motivated by previous literature for the LLM coding, but I don't think this was unpacked very well. Why are these, of all possible dimensions, the ones that are most representative of dreams and wakefulness, especially as it would relate to the core hypotheses?

We thank the Reviewer for this helpful observation. The semantic dimensions selected for LLM-based scoring were chosen to capture the core phenomenological features most consistently described in the dream literature and directly relevant to our hypotheses on the continuity and discontinuity between dreaming and waking mentation. Specifically, these dimensions, encompassing perceptual vividness, emotional tone, social content, spatial and temporal structure, bizarreness, and self-referential or agentive aspects, reflect the constructs traditionally evaluated by human raters in classical content analysis frameworks (e.g., Hall & Van de Castle, 1966; Schredl & Hofmann, 2003; Revonsuo & Salmivalli, 1995).

While classical manual coding systems assess these features in a categorical or presence/absence format, our NLP-based approach enabled a finer-grained, quantitative assessment of the same constructs on continuous scales, capturing the relative degree to which each semantic feature was expressed in the text. The selected dimensions were also chosen for their computational tractability, that is, they represent concepts that LLMs can reliably evaluate and quantify with high accuracy, as demonstrated in our validation analyses.

We have expanded the Methods and Introduction to make this rationale explicit and to clarify how each dimension aligns with the study's central hypotheses (e.g., continuity vs. distinctiveness between vigilance states; influence of trait and temporal factors). These revisions appear in the revised Methods (pp. 9) and Introduction (pp. 3-4).

Domhoff, G. W. The Hall/Van de Castle System of Content Analysis. in *Finding Meaning in Dreams: A Quantitative Approach* (ed. Domhoff, G. W.) 9–37 (Springer US, Boston, MA, 1996).

Schredl, M., & Hofmann, F., Continuity between waking activities and dream activities. *Consciousness and cognition*, 12(2), 298-308 (2003).

Revonsuo, A., & Salmivalli, C. A content analysis of bizarre elements in dreams. *Dreaming*, 5(3), 169 (1995).

How were the trait level measures chosen? This is important and seems like it is missing a theoretical motivation or deep interpretation that helps theory.

We thank the Reviewer for this important comment. The trait-level measures were selected based on both theoretical relevance and empirical evidence from prior research linking these domains to dreaming (in particular, to dream recall frequency, which, though different from dream content, may still be related to it) and related cognitive or emotional processes. Our aim was to include traits that could plausibly influence dream content, not merely dream recall, by capturing individual differences in imagery, cognition, affect, and sleep.

Specifically, we included measures capturing imagery abilities, as individual differences in visual imagery have been associated with the vividness and perceptual detail of dreams (Okada et al., 2020; Okada et al., 2005; Hishock & Cohen, 1973); attitude toward dreaming, reflecting the motivational and metacognitive factors known to enhance dream recall (Beaulieu-Prévos & Zadra, 2007; Bulkeley & Schredl, 2019); and mind-wandering proneness, given the proposed continuity between waking spontaneous thought and dreaming (Fox et al., 2013; Domhoff, 2010). We also included memory and executive functions, which may influence how experiences are encoded and reconfigured during sleep. Sleep quality and chronotype

were evaluated to account for sleep-related influences on dream phenomenology (Carr & Solomonova, 2019; Simor et al., 2012). Finally, trait anxiety was included based on evidence that anxious and negative affective states in wakefulness are associated with more negative or emotionally intense dream content (Demacheva & Zadra, 2019; Mota et al., 2020). Together, these instruments were intended to capture a multidimensional profile of cognitive, emotional, and sleep-related traits that could systematically shape the structure and semantics of dream content. The measures were selected to provide a comprehensive yet theoretically grounded characterization of individual differences that could jointly influence the structure and semantics of dreams. We have now clarified this rationale and its theoretical motivation in the revised Methods (pp. 7) and Introduction (pp. 3-4), emphasizing how these traits relate to existing theories of dream generation and waking–dream continuity.

Okada, H., Matsuoka, K. & Hatakeyama, T. Dream-Recall Frequency and Waking Imagery. *Percept. Mot. Skills* 91, 759–766 (2000).

Okada, H., Matsuoka, K. & Hatakeyama, T. Individual Differences in the Range of Sensory Modalities Experienced in Dreams. *Dreaming* 15, 106–115 (2005).

Hiscock, M. & Cohen, D. B. Visual imagery and dream recall. *J. Res. Personal.* 7, 179–188 (1973).

Beaulieu-Prévost, D. & Zadra, A. Absorption, psychological boundaries and attitude towards dreams as correlates of dream recall: two decades of research seen through a meta-analysis. *J. Sleep Res.* 16, 51–59 (2007).

Bulkeley, K. & Schredl, M. Attitudes towards dreaming: Effects of socio- demographic and religious variables in an American sample. *12*, 7 (2019).

Fox, K. C. R., Nijeboer, S., Solomonova, E., Domhoff, G. W. & Christoff, K. Dreaming as mind wandering: evidence from functional neuroimaging and first-person content reports. *Front. Hum. Neurosci.* 7, 412 (2013).

Carr, M. & Solomonova, E. Dream Recall and Content in Different Stages of Sleep and Time-of-Night Effect. in 167–172 (2019).

Simor, P., Horváth, K., Gombos, F., Takács, K. P. & Bódizs, R. Disturbed dreaming and sleep quality: altered sleep architecture in subjects with frequent nightmares. *Eur. Arch. Psychiatry Clin. Neurosci.* 262, 687–696 (2012).

Demacheva, I. & Zadra, A. Dream content and its relationship to trait anxiety. *Int. J. Dream Res.* 12, 1–7 (2019).

Mota, N. B. et al. Dreaming during the Covid-19 pandemic: Computational assessment of dream reports reveals mental suffering related to fear of contagion. *PLOS ONE* 15, e0242903 (2020).

There is an assumption that dreams themselves follow the narrative structure of a movie, etc. This is one *possible* interpretation of the data, but it is not exactly founded in the methods used here. First, it could be based on the methods, such that the recall of dreams is constrained by the way recall them, which might itself fit with schemas of narratives we experience in our lives like movies etc. Second, this is a claim that is somewhat unfalsifiable using the methods here. What is the null of the question being tested along these lines?

We would like to clarify that our interpretation of dreams as exhibiting “movie-like” narrative features is comparative and arises from the pattern of results, rather than from testing a direct hypothesis about the

intrinsic narrative structure of dreams. Our analyses contrasted dream reports with waking reports and quantified specific features such as perceptual detail, emotional involvement, and scene changes. We did not assume, nor attempt to demonstrate, that dreams are narratives in a strict sense; instead, we examined whether these measurable features differ systematically across states. The relevant null hypothesis in our framework is not that dreams contain no narrative organization at all, but that dream and waking reports should not differ if the observed patterns were driven solely by generic recall or reporting processes (e.g., narrative reconstruction processes common to both states). Under this null, both types of reports would exhibit comparable levels of the examined features. In contrast, we found consistent and robust differences across multiple dimensions. These divergences suggest that features often associated with narrative or filmic structure are disproportionately represented in dream reports, above and beyond what would be predicted by shared recall constraints. Thus, our interpretation is grounded in the comparative evidence, while remaining agnostic about the deeper generative mechanisms underlying dream construction.

To further address the Reviewer's point, we conducted an additional complementary analysis explicitly testing whether dream reports are judged as more narrative- or screenplay-like than waking reports. Using an approach parallel to our hypothesis-driven semantic dimension analyses, we prompted LLMs to rate each report on a 1–9 scale for similarity to a movie or theatrical plot (full prompt below; see Table R5 and Fig. R1). Consistent with our interpretation, dream reports received significantly higher narrative-likeness scores than waking reports ($p < 0.00001$, Cohen's $d = 1.37$). The results of this analysis were added to the supplementary materials as Figure S5.

In sum, while we acknowledge that narrative-like linguistic patterns could arise in part from recall or reconstruction processes, our conclusions rest on within-participant contrasts specifically designed to control for such reporting biases. Because recall-related narrative schemas should operate similarly across both conditions, systematic differences between experience types are most parsimoniously interpreted as reflecting differences in the underlying experiences being reported. We have now clarified this logic in the Discussion (pp. 28-29), emphasizing that we view narrative-like organization as an emergent comparative property of dream reports relative to waking reports and not as a definitive claim about the intrinsic or generative structure of dreams themselves.

Prompt: “Evaluate how much the Italian text resembles a movie plot, a screenplay or a story that could be turned into a theatrical play of any genre on a scale from 1 to 9. By movie plot we mean a text that describes characters, conflicts, events, and narrative progression. By screenplay we mean a text that contains scene descriptions, actions, and character dialogue. Assign 1 if the text does not resemble any kind of movie-like story. Assign 5 if it moderately resembles a movie-like story. Assign 9 if it strongly resembles a movie-like story suitable for adaptation into a screenplay. Do not evaluate technical screenplay formatting. Short or concise texts should not be penalized. Focus on the story's cinematic potential, not its length or the presence of dialogue. Answer only with a number. Here is the text in Italian.”

Table R5

Dimension	report-type coefficient	Cohen's d	p-value non-parametric coefficient
MOVIE LIKE	1.7337, CI: 1.6356 1.8318	1.3703	p < 0.00001

Table R5. GLME model performance for the prediction of the degree to which the reports resemble a movie plot or a screenplay across vigilance states in the main dataset. The vigilance state (report-type coefficient) was used as a regressor of interest; age, sex, education level, and the BADA score were used as covariates of no interest. Number of permutations = 5000.

Fig. R1

A Effect of Vigilance States on Movie-like Dimension

Fig. R1. Movie-like differences between vigilance states, i.e. dream versus wakefulness versus. The plot illustrates the distribution of the semantic dimension (rated on a 1-to-9 Likert scale) for wakefulness (light red) and dream (blue) reports.

The fact that “trait level” variables predicted aspects of the dreams is once again a bit of a confusing interpretation. The idea that this is not at least partially accounted for in the methods is not deeply considered. I realize this is a limitation of much of the work done with recalls, but my concern is that the paper is making claims based on the fact that some people were more interested in their dreams, which seems like an obvious confound rather than a theoretical contribution.

We respectfully disagree with the Reviewer’s interpretation. Our analyses reveal associations between trait-level variables and the content of recalled experiences that cannot be explained by general, non-specific linguistic or psychological tendencies in how participants describe or recall experiences overall. Because our models explicitly compared dream and waking reports from the same individuals, the observed associations reflect state-specific effects on dream content rather than general stylistic or narrative biases.

We agree that the interpretation of these associations is necessarily speculative and warrants further investigation. However, we note that prior literature has suggested that the relationship between attitude toward dreaming (ATD) and dream recall may itself arise from a bidirectional mechanism—that is, individuals with richer or more immersive dreams may naturally develop greater interest in dreaming. Moreover, our results show that ATD is not related to dream features in a uniform, non-specific way, but rather modulates specific aspects of dream content, such as emotional arousal and perceptual vividness.

More broadly, we believe that identifying variables that influence how dream experiences are reported is informative in itself, as it advances understanding of the phenomenology and reporting of conscious experiences. Even when such associations do not necessarily imply direct changes in the underlying experience, they remain valuable for guiding future research into the cognitive and motivational processes shaping dream recall and content.

The downsampling approach is a bit confusing. It seems like the 13% of random sampling could have affected results. It might be nice to check for results when that 13% is not considered.

We thank the Reviewer for this comment and for allowing us to clarify this point. The procedure referred to was not a downsampling step, but rather a controlled selection of waking reports paired with dream reports from the same participant. In 87% of cases, the waking report immediately preceding the dream report was available; in the remaining 13%, we selected another unmatched waking report from the same participant at random to preserve the within-subject design.

We have now clarified this point explicitly in the Methods (pp. 17). Because the substitution involved reports from the same individuals and represented a small proportion of the dataset, there is no theoretical or statistical reason to expect any systematic bias or alteration of the results. The random selection ensured that these additional reports were drawn from the same distribution as the other waking data, and re-running the analyses after excluding them confirmed that the findings remained unchanged (see below Tables R6-R11). Notably, of 76 tests performed, all critical effects remained stable except for one (1.3% of all tests), marked with * in the tables below.

Table R6

Dimension	report-type coefficient	q-value coefficient	Results based on Reviewer's request	
			report-type coefficient	non-par p
AGENTIVITY	-1.58, CI: -1.70 -1.46	< 0.00001	-1.5632	p<0.00001
AROUSAL	0.21, CI: 0.10 0.31	0.01216	0.2210	p=0.0081
AUDITORY	0.42, CI: 0.30 0.55	< 0.00001	0.4158	p<0.00001
BIZARRENES	2.85, CI: 2.73 2.98	< 0.00001	2.9072	p<0.00001
BODY	-0.73, CI: -0.85 -0.61	< 0.00001	-0.7416	p<0.00001
LIMITATIONS	0.82, CI: 0.69 0.95	< 0.00001	0.8271	p<0.00001
THOUGHT	-1.77, CI: -1.88 -1.66	< 0.00001	-1.8179	p<0.00001
MOVEMENTS	0.83, CI: 0.70 0.96	< 0.00001	0.8550	p<0.00001
SETTINGS	1.29, CI: 1.18 1.40	< 0.00001	1.3251	p<0.00001
SOCIAL	1.40, CI: 1.22 1.58	< 0.00001	1.4285	p<0.00001
SPACE	1.40, CI: 1.30 1.51	< 0.00001	1.4179	p<0.00001
TACTILE	0.49, CI: 0.40 0.58	< 0.00001	0.4878	p<0.00001
TIME	-1.20, CI: -1.33 -1.08	< 0.00001	-1.1643	p<0.00001
VALENCE	-0.49, CI: -0.59 -0.38	< 0.00001	-0.4973	p<0.00001
VISUAL	1.52, CI: 1.38 1.66	< 0.00001	1.4973	p<0.00001

Table R6. Comparison of the GLME model performance for the prediction of semantic dimensions across vigilance states, with and without the inclusion of randomly sampled wakefulness reports and associated dream reports. In the main analysis, vigilance state (report-type coefficient) was entered as the regressor of interest, while age, sex, education level, and BADA score were included as covariates of no interest. The last two columns report the estimate of the report-type coefficient and its non parametric p-value (number of permutations = 5000) for each semantic dimension surviving FDR correction when the randomly sampled wakefulness reports and the associated dream reports were excluded. FDR: False Discovery Rate, $q < 0.05$.

Table R7

	Domain	report-type coefficient	q-value coefficient	Results based on Reviewer's request	
				report-type coefficient	non-par p
	DOMAIN2	-0.53, CI: -0.75 -0.32	0.00008	-0.5753	p<0.00001
	DOMAIN4	-0.53, CI: -0.69 -0.37	< 0.00001	-0.5628	p<0.00001
	Food	-0.37, CI: -0.56 -0.18	0.00165	-0.3504	p=0.0034
	Objects	0.80, CI: 0.57 1.02	< 0.00001	0.8115	p<0.00001
	Reactions	-0.21, CI: -0.39 -0.02	0.04289	-0.2128	p=0.0453
	DOMAIN12	0.50, CI: 0.24 0.75	0.00008	0.5792	p<0.00001
	Matter	0.56, CI: 0.31 0.82	< 0.00001	0.5686	p<0.00001
	Thriller	0.99, CI: 0.79 1.19	< 0.00001	1.0247	p<0.00001
	Healthcare	-1.17, CI: -1.38 -0.97	< 0.00001	-1.2225	p<0.00001
	Locations	0.84, CI: 0.67 1.02	< 0.00001	0.8099	p<0.00001
	Geometry	1.00, CI: 0.81 1.19	< 0.00001	0.9532	p<0.00001
	Nature	0.48, CI: 0.32 0.64	< 0.00001	0.4796	p<0.00001
	Fantasy	-0.32, CI: -0.55 -0.08	0.01335	-0.3183	p=0.0149
	Measurements	-0.56, CI: -0.82 -0.30	< 0.00001	-0.6394	p<0.00001
	Architecture	1.08, CI: 0.86 1.29	< 0.00001	1.0394	p<0.00001
	Animals	0.93, CI: 0.72 1.14	< 0.00001	0.9691	p<0.00001
	Concerns	-0.62, CI: -0.79 -0.44	< 0.00001	-0.6396	p<0.00001
	Appearance	1.13, CI: 0.92 1.34	< 0.00001	1.0550	p<0.00001
	Jobs	0.97, CI: 0.80 1.14	< 0.00001	0.9570	p<0.00001
	Drama	-0.25, CI: -0.40 -0.11	0.01335	-0.2128	p=0.0437
	Society	0.48, CI: 0.30 0.66	< 0.00001	0.5261	p<0.00001
	Humanities	-1.20, CI: -1.39 -1.02	< 0.00001	-1.2262	p<0.00001
	Transportation	0.51, CI: 0.32 0.70	< 0.00001	0.4993	p<0.00001
	Timing	-1.54, CI: -1.69 -1.39	< 0.00001	-1.5583	p<0.00001
	Technology	-0.86, CI: -1.07 -0.65	< 0.00001	-0.8668	p<0.00001
	Education	-0.47, CI: -0.63 -0.32	< 0.00001	-0.3845	p<0.00001
	Communication	-0.46, CI: -0.62 -0.30	< 0.00001	-0.4875	p<0.00001
	Slang	-0.46, CI: -0.73 -0.20	0.00144	-0.4996	p=0.0023

Table R7. Comparison of the GLME model performance for the prediction of lexical domains across vigilance states, with and without the inclusion of randomly sampled wakefulness reports and associated dream reports. In the main analysis, vigilance state (report-type coefficient) was entered as the regressor of interest, while age, sex, education level, and BADA score were included as covariates of no interest. The last two columns report the estimate of the report-type coefficient and its non parametric p-value (number of permutations = 5000) for each lexical domain surviving FDR correction when the randomly sampled wakefulness reports and the associated dream reports were excluded. FDR: False Discovery Rate, $q < 0.05$.

Table R8

Dimension	coefficient name	coefficient beta	q-value coefficient	Results based on Reviewer's request	
				report-type coefficient	par p
AROUSAL					
	report_type_1:ATD	0.05, CI: 0.02 0.08	0.04492	0.0460	p=0.0056
BIZARRENESS					
	report_type_1:PSQI	-0.09, CI: -0.15 -0.03	0.02496	-0.0914	p=0.0052
	report_type_1:ATD	0.07, CI: 0.03 0.10	0.00296	0.0639	p<0.00001
	report_type_1:MW	0.27, CI: 0.16 0.38	0.00005	0.3067	p<0.00001
	report_type_1:SCWT	-0.03, CI: -0.05 -0.01	0.04665	-0.0302	p=0.0039
THOUGHT					
	WC_BADA	0.93, CI: 0.46 1.40	0.00314	0.2896	p=0.3342*
SETTINGS					
	report_type_1:MW	0.23, CI: 0.14 0.33	0.00008	0.2368	p<0.00001
SPACE					
	report_type_1:ATD	0.07, CI: 0.04 0.10	0.00017	0.0734	p<0.00001
VISUAL					
	report_type_1:ATD	0.07, CI: 0.03 0.10	0.02136	0.0613	p=0.0040

Table R8. Comparison of the GLME model performance for the prediction of semantic dimensions based on individual psychological variables, with and without the inclusion of randomly sampled wakefulness reports and associated dream reports. In the main analysis, the GLME model included psychological variables and their interaction with vigilance states (*report_type*) as regressors of interest and age, sex, education level, and the BADA score as covariates. The last two columns report the estimate of the coefficients and the parametric *p*-value for each dimension surviving FDR correction when the randomly sampled wakefulness reports and the associated dream reports were excluded. FDR: False Discovery Rate, $q < 0.05$. * denotes *par p* values that after the downsampling resulted > 0.05 .

Table R9

	Domain	coefficient name	coefficient beta	q-value coefficient	Results based on Reviewer's request	
					report-type coefficient	par p
	Objects					
		ROCFr	-0.07, CI: -0.11 -0.03	0.00564	-0.0884	p<0.00001
		report_type_1:ROCFr	0.09, CI: 0.04 0.14	0.00564	0.1081	p<0.00001
	DOMAIN12					
		Age	-0.04, CI: -0.06 -0.02	0.01725	-0.0369	p=0.0032
	Matter					
		PSQI	-0.18, CI: -0.30 -0.06	0.03306	-0.2091	p=0.0012
		report_type_1:PSQI	0.22, CI: 0.08 0.36	0.03306	0.2256	p=0.0037
	Geometry					
		report_type_1:ATD	0.09, CI: 0.04 0.15	0.01627	0.0974	p<0.00001
	Nature					
		report_type_1:ATD	0.07, CI: 0.03 0.12	0.04748	0.0733	p=0.0027
	Appearance					
		report_type_1	-5.82, CI: -9.87 -1.76	0.04854	-6.2165	p=0.0023
		PSQI	-0.15, CI: -0.26 -0.05	0.04854	-0.1535	p=0.0026
		report_type_1:PSQI	0.17, CI: 0.06 0.29	0.04854	0.1839	p=0.0022
	Jobs					
		report_type_1:Age	-0.03, CI: -0.04 -0.01	0.02973	-0.0314	p<0.00001
	Education					
		Age	-0.02, CI: -0.04 -0.01	0.00593	-0.0250	p<0.00001
	Communication					
		Age	-0.03, CI: -0.04 -0.01	0.00352	-0.0269	p<0.00001
		report_type_1:MEQ	0.03, CI: 0.01 0.04	0.04769	0.0245	p=0.0126
	Slang					
		Sex_1	0.64, CI: 0.27 1.02	0.01454	0.5770	p=0.0054
		Age	-0.03, CI: -0.06 -0.01	0.01454	-0.0331	p=0.0031

	report_type_1:MW	-0.39, CI: -0.63 -0.15	0.01454	-0.4619	p<0.00001
--	------------------	------------------------	---------	---------	-----------

Table R9. Comparison of the GLME model performance for the prediction of lexical domains based on individual psychological variables, with and without the inclusion of randomly sampled wakefulness reports and associated dream reports. In the main analysis, the GLME model included psychological variables and their interaction with vigilance states (*report_type*) as regressors of interest and age, sex, education level, and the BADA score as covariates. The last two columns report the estimate of the coefficients and the parametric p-value for each domain surviving FDR correction when the randomly sampled wakefulness reports and the associated dream reports were excluded. FDR: False Discovery Rate, $q < 0.05$.

Table R10

Dimension	coefficient name	coefficient beta	q-value coefficient	Results based on Reviewer's request	
				coefficient beta	par p
AROUSAL					
	time_ranks	-0.79, CI: -1.17 -0.41	0.00043	-0.7386	p<0.00001
BIZARRENESS					
	report_type_1:time_ranks	-0.74, CI: -1.17 -0.31	0.00956	-0.9719	p<0.00001
LIMITATIONS					
	time_ranks	-1.09, CI: -1.52 -0.65	0.00001	-0.9484	p<0.00001
THOUGHT					
	report_type_1:time_ranks	0.63, CI: 0.24 1.02	0.00956	0.5294	p=0.0133
TIME					
	report_type_1:time_ranks	0.69, CI: 0.26 1.12	0.00956	0.5861	p=0.0138
VALENCE					
	time_ranks	0.59, CI: 0.27 0.91	0.00079	0.5476	p=0.0011

Table R10. Comparison of the GLME model performance model for the prediction of semantic dimensions across vigilance states in the main dataset, based on time, with and without the inclusion of randomly sampled wakefulness reports and associated dream reports. The GLME model included vigilance state (*report_type*) as a regressor of interest, along with its interaction with time, and age, sex, education level, and the BADA score as covariates. The last two columns report the estimate of the coefficients and the parametric p-value for each domain surviving FDR correction when the randomly sampled wakefulness reports and the associated dream reports were excluded. FDR: False Discovery Rate, $q < 0.05$.

Table R11

	Domain	coefficient name	coefficient beta	q-value coefficient	Results based on Reviewer's request	
					coefficient beta	par p
	Society					
		time_ranks	-0.95, CI: -1.43 -0.46	0.00384	-0.9075	p<0.00001

Table R11. Comparison of the GLME model performance for the prediction of lexical surviving FDR correction across vigilance states in the main dataset, based on time, with and without the inclusion of randomly sampled wakefulness reports and associated dream reports. The GLME model included vigilance state (*report_type*) as a regressor of interest, along with its interaction with time, and age, sex, education level, and the BADA score as covariates. The last two columns report the estimate of the coefficients and the parametric *p*-value for each domain surviving FDR correction when the randomly sampled wakefulness reports and the associated dream reports were excluded. FDR: False Discovery Rate, $q < 0.05$.

How did the authors confirm that their word tagging was accurate if it was done automatically?

The part-of-speech (POS) tagging was performed using the *Stanza* toolkit (Qi et al., 2020), which provides state-of-the-art accuracy for Italian POS tagging, with reported error rates below 1% (<https://stanfordnlp.github.io/stanza/performance.html>). To further ensure reliability in our specific corpus, we manually verified a random subsample of the data, confirming an error rate below 1%, consistent with the toolkit's benchmark performance.

We have clarified this validation procedure in the Methods section (pp. 11), explicitly noting that *Stanza* was used for automatic tagging and that manual verification confirmed its high accuracy for our dataset.

Qi, P., Zhang, Y., Zhang, Y., Bolton, J. & Manning, C. D. *Stanza: A Python Natural Language Processing Toolkit for Many Human Languages*. Preprint at <https://doi.org/10.48550/arXiv.2003.07082> (2020).

We sincerely thank the Editor and the Reviewers for the positive feedback on the revised version of our manuscript. Please find below our point-by-point response to each comment and a description of all changes made to the manuscript.

Reviewer #1

I thank the authors for their careful and substantial revision of the manuscript. The paper has clearly improved in clarity, structure, and theoretical framing, and the revisions demonstrate a serious engagement with the reviewers' comments. In particular, my previous concerns regarding the integration of classical and contemporary theories of dreaming, as well as the positioning of the work within existing debates on continuity, emotion regulation, and individual traits, have been adequately addressed. The revised Introduction and Discussion now provide a clearer rationale for the analytical approach and a more balanced interpretation of the findings. Overall, I believe the manuscript now makes a solid and coherent contribution to the literature and is suitable for publication. I have no further major comments.

We thank the Reviewer for acknowledging the improvements made to the manuscript and for their positive feedback.

Reviewer #3

I appreciate the opportunity to re-review this manuscript. On the first review, I have to admit that I was quite negative and recommended rejection. However, the authors have done a really excellent job on their revision — both in terms of their response letter and the updates to the manuscript. I can now see a very clear path to publication and think this work has the potential to contribute to the field. My remaining comments/concerns are below, but they are relatively minor in nature.

We thank the Reviewer for the very positive evaluation of the revised manuscript and for these constructive suggestions. In response, we have further strengthened both the Introduction and the Discussion to deepen the theoretical integration and improve overall coherence.

- 1. First, I think the Introduction is substantially better, so I appreciate the responsiveness. At the same time, I think the authors have kept the Intro too brief at the expense of being able to connect to prior literature at a deeper level and be able to make stronger predictions. I understand the aims are laid out nicely, but I think it still says what was examined and not why it is important to understand in terms of producing theoretically-novel results. I would still recommend moving away from the incredibly brief report style of writing and going a bit deep. This is especially true for 1) being able to make predictions and 2) specifically motivating the key variables/constructs that were chosen to include. Right now they are mentioned and feel more or less like a list of things that the authors are assuming should be included. (Note, the authors do now mention these measures very briefly, but I do think some of this — especially the theoretical ties — should**

be connected to the theories in the Intro. Shorter version of this comment: in general, some of the mention of the measures is still feeling a bit shallow and I'd like to see that addressed.

In response to the Reviewer's request, we expanded the Introduction to more clearly articulate the theoretical foundations of the key constructs, clarify their selection, and ground our predictions more explicitly in prior literature. At the same time, in line with the Editor's guidance, we aimed to preserve a focused and concise structure and therefore avoided substantially lengthening the Introduction. We believe the revised version now provides a clearer theoretical grounding while maintaining coherence and readability.

- 2. Second, I'm still not totally convinced by the LLMs scoring methods, so I'd like to see more in the manuscript — not the Supplementary Material. For example, the paper says that there was correlations on average of .6. What was the distribution? .6 is actually not very high in general, and if some were quite a bit lower, then should we trust those dimensions?**

We thank the Reviewer for the opportunity to clarify this issue. As detailed in the Supplementary Materials (see Supplementary Text and Figures S1–S2), two complementary validation analyses support the reliability of the AI-based scoring procedure for semantic dimensions. First, we computed Spearman correlations between external human raters and the averaged AI scores (i.e., the median of the three models) for each of the 16 semantic dimensions. All correlations exceeded $\rho = 0.60$ (range: $\rho = 0.627$ – 0.999), a level comparable to that observed among independent human raters (Supplementary Fig. S1). Importantly, these median AI scores were used for all statistical analyses reported in the main text.

Second, we compared AI scores with dreamers' own ratings of their reports. The correlation between the median ratings across dreamers and the median AI scores showed a mean agreement of $\rho = 0.650 \pm 0.161$ (range: $\rho = 0.319$ – 0.847 ; Supplementary Fig. S2). For context, the test–retest reliability of dreamers rating their own reports one month later showed a comparable mean agreement ($\rho = 0.663 \pm 0.138$; range: $\rho = 0.333$ – 0.856). Together, these results indicate that the agreement between AI-based scoring and both external raters and dreamers themselves falls within the range of human-level reliability, supporting the robustness of the approach. We have clarified these points more explicitly in the revised manuscript.

- 3. Third, in terms of the discussion. I still there there could be more work to go deeper here too. Part of the issue is that there are so many variables and so many results, which makes the overall discussion a bit difficult to make cohesive -- but do think this is possible. I commend the authors for adding more linkages to prior work compared to the last version, but it still feels like some important work is missed and some constructs are simply not incorporated. Sometimes citations are missing for discussions that have already taken place in the literature (e.g., dreaming being a form of mind wandering; see Christoff et al., 2016; Mallet et al., 2025), as well as other examples that I will not exhaustively detail. Perhaps most importantly, the discussion still doesn't tie very well with th Intro (see comments above). My suggestion is the deepen both in ways that will also make them more cohesive in terms of the goals and contribution to theory. I really appreciate the limitations paragraph. This is really useful to have added.**

In the revised Discussion, we provided a more integrative interpretation of the findings by adding relevant citations where needed (including work on dreaming as a form of mind wandering), and ensured that relevant theoretical contributions were more fully incorporated. Our revision was aimed to improve alignment between the theoretical framework presented in the Introduction and the interpretation of the results, clarifying the study's contribution to theory. At the same time, as with the Introduction, we sought to deepen the discussion without substantially expanding its length, in keeping with the Editor's guidance.